# Radioprotection and Radiomitigation: From the Bench to Clinical Practice

**DOI:** 10.3390/biomedicines8110461

**Published:** 2020-10-30

**Authors:** Elena Obrador, Rosario Salvador, Juan I. Villaescusa, José M. Soriano, José M. Estrela, Alegría Montoro

**Affiliations:** 1Department of Physiology, Faculty of Medicine and Odontology, University of Valencia, 46010 Valencia, Spain; elena.obrador@uv.es (E.O.); rosario.salvador@uv.es (R.S.); Jose.M.Estrela@uv.es (J.M.E.); 2Service of Radiological Protection, Clinical Area of Medical Image, La Fe University Hospital, 46026 Valencia, Spain; villaescusa_ign@gva.es; 3Biomedical Imaging Research Group GIBI230, Health Research Institute (IISLaFe), La Fe University Hospital, 46026 Valencia, Spain; 4Food & Health Lab, Institute of Materials Science, University of Valencia, 46980 Valencia, Spain; Jose.Soriano@uv.es; 5Joint Research Unit in Endocrinology, Nutrition and Clinical Dietetics, University of Valencia-Health Research Institute IISLaFe, 46026 Valencia, Spain

**Keywords:** ionizing radiations, radioprotectors, radiomitigators, free radicals, antioxidants

## Abstract

The development of protective agents against harmful radiations has been a subject of investigation for decades. However, effective (ideal) radioprotectors and radiomitigators remain an unsolved problem. Because ionizing radiation-induced cellular damage is primarily attributed to free radicals, radical scavengers are promising as potential radioprotectors. Early development of such agents focused on thiol synthetic compounds, e.g., amifostine (2-(3-aminopropylamino) ethylsulfanylphosphonic acid), approved as a radioprotector by the Food and Drug Administration (FDA, USA) but for limited clinical indications and not for nonclinical uses. To date, no new chemical entity has been approved by the FDA as a radiation countermeasure for acute radiation syndrome (ARS). All FDA-approved radiation countermeasures (filgrastim, a recombinant DNA form of the naturally occurring granulocyte colony-stimulating factor, G-CSF; pegfilgrastim, a PEGylated form of the recombinant human G-CSF; sargramostim, a recombinant granulocyte macrophage colony-stimulating factor, GM-CSF) are classified as radiomitigators. No radioprotector that can be administered prior to exposure has been approved for ARS. This differentiates radioprotectors (reduce direct damage caused by radiation) and radiomitigators (minimize toxicity even after radiation has been delivered). Molecules under development with the aim of reaching clinical practice and other nonclinical applications are discussed. Assays to evaluate the biological effects of ionizing radiations are also analyzed.

## 1. Introduction

Radiation is defined as energy emission or energy transmission via some type of medium, either as waves or as subatomic particles. Ionizing radiation is the energy released by atoms in the form of electromagnetic waves (e.g., X or gamma rays) or particle radiation (alpha, beta, electrons, protons, neutrons, mesons, prions, and heavy ions) with sufficient energy to ionize atoms or molecules. Ionizing radiation emission can occur as a consequence of the decay process of unstable nuclei or due to nuclear de-excitation in devices such as nuclear reactors, X-ray machines, and cyclotrons. Radioactivity is defined as spontaneous disintegration of atoms. The excess energy emitted in this process is considered as a type of ionizing radiation. Unstable elements that disintegrate in this process and emit ionizing radiation are called radionuclides. The activity of a radionuclide is expressed in becquerels (one Bq is one disintegration per second) [1].

The absorption of radiation-derived energy by biological materials may cause excitation or ionization. Sufficient energy can cause the ejection of one or more orbital electrons from an atom or molecule, a process known as ionization, and such radiation is called ionizing radiation [2].

Radiation-induced biological effects are determined by factors such as dose rate, total dose, linear energy transfer (LET), and total doses fractionation and protraction. They are also determined by other factors such as repair mechanisms, bystander effects (nonirradiated cells respond to signals received from nearby irradiated cells), and exposure to chemical carcinogens, tumor promoters, and other toxins [3].

Around 50% of radiation exposure for humans occurs from natural sources. This includes cosmic rays and terrestrial sources, which include about 60 naturally occurring radioactive materials. These materials are abundant in the earth’s crust, as well as in the air, food and water, and even the human body [4]. Other significant sources producing radiation exposure in the population are manmade. This comprises a range of activities including radiation and radioisotopes used in healthcare, occupational sources resulting from the process of electricity generation from nuclear power reactors, application of nuclear techniques in industries, nuclear weapons testing, wars, or even terrorism. The use of ionizing radiation in medical diagnosis and therapy is widespread and constantly increasing due to novel health-related applications. In total, 98% of the population dose contribution from all artificial sources is a result of medical radiation [5]. Current multidisciplinary research in the field of radioprotection and radiomitigation involves all aspects of basic and clinical research, i.e., radiological procedures in diagnostic radiology, image-guided interventional procedures, nuclear medicine, and radiotherapy. The collective effective dose resulting from medical radiation has three main sources: (1) computed tomography (CT) scans, (2) nuclear medicine (including nuclear cardiology), and (3) interventional radiology and cardiology [6,7]. The contribution of radiotherapy to the collective effective dose received by the general population has been a matter of controversy. For instance, the National Council on Radiation Protection (NCRP, USA) Report No. 184 committee elected to not incorporate radiation dose from radiotherapy into its calculated population dose exposures [8]. The basic argument raised in this report was that assessment of the collective effective dose for the population undergoing radiotherapy is more complex than that for other medical radiation exposures. However, the large number of patients receiving courses of radiotherapy, which frequently include different types of imaging, raises the question of whether radiotherapy-derived doses should also be considered for calculations. Nevertheless, in most cases, radiotherapy is applied to an organ or specific part of the body. Moreover, heterogeneity in radiotherapy regarding the individual dose distribution adds complexity to the actual contribution to the collective effective dose. These facts and the limited number of patients receiving radiotherapy, compared to those subjected to complex radiology or computed axial tomography, may explain the position expressed in the above mentioned NCRP report.

There are two types of radiation exposure: internal and external. If a radionuclide is inhaled, ingested, or enters in the bloodstream by any other means, it is called an internal exposure. On the other hand, the deposition of airborne radioactive material (e.g., dust, liquid, or aerosols) on the skin or garment constitutes an external exposure [9]. Potential biological effects and damage caused by radiation depend on the variety and dosage of the received radiation, as well as the exposure conditions. Adverse effects may be acute (lasting a few days or weeks after radiation exposure), intermediate (weeks to months), or late (months to years) [10,11].

All this has produced the need to develop effective countermeasures to achieve protection against harmful radiation. Medical countermeasures (MCM) is a term adopted by the United States (US) Departments of Defense and Health and Human Services and refers to any type of agent used for prevention (radioprotectors and radiomitigators) or treatment (therapeutics) of radiation injuries [12,13].

Radioprotectors are compounds capable of reducing the damage induced by radiation in normal tissues. Radiomitigators are compounds that can diminish toxicity, even once the radiation is delivered. Could a potential radiomitigator also be classified as preventive? That would depend on whether its administration before radiation exposure could offer benefits. For instance, this would be true for a molecule capable of upregulating the DNA repair mechanisms when given before damaging irradiation. In general, ideal radioprotectors or radiomitigators should be stable, offer the possibility of easy administration, have no relevant toxicity, and protect normal tissues that are considered sensitive, in which acute or late toxicity would be either dose-limiting or responsible for a significant reduction in quality of life (causing, e.g., mucositis, pneumonitis, myelopathy, xerostomia, proctitis, fibrosis, and leukoencephalopathy) [14]. However, despite many years of study, an ideal radioprotector and/or radiomitigator has not yet been found.

## 2. Interaction of Ionizing Radiation with Living Matter

### 2.1. The Effects Are Determined by the Radiation Type and Its Penetration Capacity

Distribution of ionization and excitation along the track of an ionizing particle will vary according to the type of radiation. A useful comparative term to describe the deposition of energy by different types of radiation is linear energy transfer (LET) or the amount of energy that a specific ionizing particle transfers to the material traversed per unit distance. Thus, LET directly affects the relative biological effectiveness (RBE) of a specific radiation type [2]. The RBE is defined as the ratio between an absorbed standard dose of radiation (typically X) and the absorbed dose of any other type of radiation that causes the same amount of biological damage. In many cases, the biological effect of radiation increases in proportion to the increase in LET. Radiations commonly used to assess RBE are low-LET X or γ, for which RBE is 1.0. However, when evaluating some biological effects caused by high-LET radiation (such as fast neutrons), the RBE can vary widely, from about 3 to more than 100 depending on the cellular or tissue effect considered. For example, higher RBEs for neutron radiation are associated with high LET effects, which are directly linked with protons released by collisions of these neutrons with hydrogen nuclei [15]. Consequently, doses should be evaluated in terms of absorbed dose (in Grays, Gy), and, when high-LET radiations are involved, the absorbed dose must be correlated with an appropriate RBE. The RBE is correlated with the amount of the radiation dose absorbed, expressed in Gy (1 Gy is 1 joule of radiation energy absorbed per kg of matter). In parallel, the Sievert (Sv), defined as the corresponding biological effect of the deposit of one joule of radiation energy in 1 kg of human tissue (www.icrp.org), is used to evaluate the biological effect of low doses of ionizing radiation representing the risk of external radiation from sources outside the body, as well as those representing the risk of internal irradiation due to accidentally inhaled or ingested radioactive substances. The Sv helps to value the stochastic health risk, which represents the probability of radiation-induced cancer and genetic damages. The following considerations describe the most common types of radiation:

Alpha radiation happens when an atom goes through radioactive decay, emanating a particle composed of two protons and two neutrons (e.g., a helium-4 atom’s nucleus). α particles interact heavily with matter because of their charge and mass. They, however, travel merely a few centimeters through the air. Thus, they cannot enter into the external layer of dead skin cells. However, a substance emitting α can be very deleterious for the cell, in cases where it is ingested through food or air [16].

Beta radiation may be either an electron or a positron. Because of having lower mass, this radiation can travel a few meters in the air, but some dense pieces of plastic or a pile of paper can block it. This type can enter into the skin a few centimeters deep. However, its main threat lies in internal emissions caused by ingested material [17].

Gamma radiation entails an emission of a photon of energy from an unstable nucleus. γ radiation is capable of traveling much longer distances through the air because it has no mass or charge; in every 150 m, it loses approximately half its energy. γ radiation can be blocked by a thick or dense enough material, e.g., lead or depleted uranium. X-rays behave in an analogues manner to γ radiation; however, compared to γ radiation, their wavelength is longer and (usually) their energy is lower. X-rays originate from energy changes in an electron, such as moving from a higher energy level to a lower one, which causes the release of excess energy [18].

Neutron radiation is composed of free neutrons, resulting from nuclear fusion, which could be spontaneous or induced. They are capable of traveling hundreds to thousands of meters in the air; a hydrogen-rich material, such as concrete or water, can block them. Neutrons have no charge and cannot ionize an atom directly. Thus, when they are absorbed into a stable atom, they commonly cause indirect ionization. This makes them unstable and consequently emit other types of ionizing radiation [19]. After neutrons strike the hydrogen nuclei, proton radiation (fast protons) is produced. These protons that have high energy, are charged, and interact with the electrons in matter are considered ionizing particles [20].

Proton and carbon ion therapy are two types of hadron therapy which have been increasingly used in recent years for cancer treatment. Proton therapy uses a beam of protons to irradiate tissues, most often as a type of cancer therapy. Its main advantage is that the dose is deposited over a narrow range of depth, which results in minimal entry, exit, or scattered radiation dose to healthy nearby tissues [21]. Carbon ions exhibit a characteristic energy distribution in depth, known as the “Bragg peak,” where low levels of energy are deposited in tissues proximal to the target, and the majority of energy is released in the target. Its main advantage is that it may allow dose escalation to tumors while reducing radiation dose to adjacent normal tissues [22].

### 2.2. Physical, Biological, and Chemical Regulatory Factors

Many different physical, biological, and chemical regulatory factors influence the effects of radiation. Physical regulation implies that the kinetic energy of radiation is transferred to atoms or molecules, thus leading to their excitation and ionization. This is influenced by the time, dose and dose rate, fractionation regimen, volume of tissue irradiated, temperature, and type of radiation. Biological regulators relate to the type of cell/tissue/organ, its sensitivity, bystander effects, age, and physiological mechanisms of reparation. Chemical regulators include radiosensitizers, radioprotectors, radiomitigators, and therapeutic agents [23,24]. The effect of a specific type of radiation in living matter will depend on a combination of these factors. Consequently, the development of a therapy to prevent or treat damage from radiation must take into account the relative influence of different regulatory factors.

### 2.3. Exposure to Ionizing Radiation: Adverse Effects

Most adverse effects of exposure to ionizing radiation can be assigned to two types of categories. The first is deterministic or predictable (in a time range known a posteriori of the event) and due to harmful tissue/organ damage following high doses of radiation; as a function of the time interval between irradiation and its observable effect, deterministic effects may be classified as early or late effects. The second is stochastic (random), i.e., cell mutation-associated pathologies (mainly cancer) and heritable effects following moderate and possibly low doses [25]. Thus, it seems that following the well-established radiobiological concept of no radiation dose can be considered completely safe is judicious [26].

Acute effects are generated due to the death of considerable number of cells in tissues that have rapid turnover rates (e.g., the bone marrow, epidermis, and mucosae of the upper and lower intestinal tract). The effects are usually revealed in a timespan of days to even weeks after the irradiation [27]. This response is usually associated with inflammation, which might be directly produced by the radiation exposure or secondary to cell loss [28,29]. The local release of proinflammatory factors (e.g., IL-1, TNF-α, COX-2, NO) can trigger the proliferation of damaging radicals, e.g., reactive oxygen species (ROS), on top of the radicals directly produced by the ionizing radiation [30].

Late responses tend to occur in tissues with slow cell turnover. They are normally persistent and progressive, located in organs with infrequent parenchymal cell division (such as the kidney and liver) or those that do not divide (for example, the central nervous system (CNS) and muscles) [31]. Their nature and timing are dependent upon the involved tissue and could manifest as a decrease in organ function—for instance, radiation-induced nephropathy—thus causing hypertension, high creatinine, elevated blood nitrogen levels, and functional loss [32]. Another common example is the development of tissue fibrosis (increased collagen synthesis and deposition) that happens in a range of tissues (including subcutaneous tissue, muscle, lung, and the gastrointestinal tract), sometimes a few years after irradiation [33]. It seems that fibrosis is associated with the abnormal and chronic expression of proinflammatory cytokines. The immune system (i.e., macrophages and mast cells) contributes to the fibrotic reaction [34]. Moreover, connective tissue damage and/or an impairment in the vasculature of an organ potentially cause a progressive impairment of organ circulation. Under these circumstances, secondary cell death may occur due to nutrient deprivation [35]. Stochastic effects likely derive from an injury to a single cell or a small number of cells. Cancer induction is the most important somatic late effect of low-dose radiation exposure. Figure 1 schematically describes the main consequences of exposure to ionizing radiation in organs and tissues.

### 2.4. Acute and Chronic Radiation Syndromes

Acute radiation syndrome (ARS) involves a number of health effects caused by exposure to elevated amounts of ionizing radiation (total dose >0.7 Gy) in a limited timespan [36]. The symptomatology might begin less than one hour after the dose is received and last for months [37]. The prodromal stage is accompanied by general symptoms such as headaches, vertigo, muscle weakness, and abnormal sensations of taste or smell. Any exposure to 1–2 Gy leads to NVD (i.e., nausea, vomiting, diarrhea) as part of ARS-related prodromal stage. However, exposure to 2–6 Gy produces hematopoietic syndrome, which affects the bone marrow, spleen, and thymus. An exposure to 8–15 Gy could produce gastrointestinal syndrome, while exposure to >25 Gys can provoke CNS syndrome [37]. NVD might be associated with flu-like symptoms of fever and/or faintness. Such symptoms, however, tend to attenuate rapidly, and the patient might mistakenly have a sense of being recovered because of degeneration and repair of proliferative tissues. These symptoms represent a latent stage. On the basis of the received dose, the latent stage could continue for a few hours or even up to a few days. Later in time, and as the dose received increases, more severe damage may occur, affecting the skin (reddening, blistering, and/or ulceration), lungs (inflammation), bone marrow (leukopenia, thrombopenia, and increased sedimentation rate), gastrointestinal tract (inflammation and/or bleeding), cardiovascular system (arrhythmia, fall of blood pressure), and CNS (increased irritability, insomnia, fear, and symptoms derived from damage affecting neuromotor functions). This damage may eventually cause death. ARS is generally a rare event, although it may affect a large number of people in the case of an accident such as that in Chernobyl. ARS treatment may require blood transfusions, antibiotics, colony-stimulating factors, or even stem-cell transplants [37,38].

Chronic radiation syndrome (CRS) involves radiation-induced health effects that may require years to develop after exposure. The threshold for CRS is around 0.7 and 1.5 Gy, at dose rates >0.1 Gy/year, and cumulative doses exceeding 2–3 Gy over 2–3 years [39]. Its latency can comprise a period of 1–5 years and has been defined as a systemic response of the body to chronic total body exposure in humans. The early symptomatology of CRS can involve alterations in vegetative functions and, eventually, changes in tactile and olfactory sensitivity. At a more advanced stage, gastrointestinal toxicity (transmural injury of the bowel wall might later cause progressive vasculitis, thrombosis, and, finally, variable grades of ischemia and necrosis), atrophy of the skin and muscle, and eye cataracts are common. Genetic damage-related cancer, i.e., different solid cancers or leukemia, may also develop either at an early stage or later in time [40,41].

## 3. Radioprotectors, Radiomitigators, and Other Antiradiation Therapies

This section describes many compounds. Due to their diversity, we grouped them by their chemical origins and/or more general functions. This discussion focuses only on molecules showing promising properties under in vivo conditions.

### 3.1. Radioprotectors

#### 3.1.1. Thiol-Containing Molecules

The potential role of sulfur compounds as radioprotectors has been known for a long time, when it was first observed that exogenous thiols, if present at the moment of irradiation, could protect cells in vitro [42] and also in experimental animals [43]. Since then, many thiol-containing compounds have been tested in vitro and in vivo. Nevertheless, it has been noted that thiols may have less of an effect on aerobic radiosensitive cells or hypoxic radioresistant cells [44]. At present, the only thiol approved by the FDA as a cytoprotective adjuvant is amifostine (WR-2721). Nevertheless, this approval is limited to reducing the nephrotoxic effects caused by repeated treatment with cisplatin for advanced ovarian cancer and to attenuate mouth dryness resulting from radiation therapy after surgery in individuals suffering from head and neck cancer [45]. For instance, a clinical trial on the evaluation of amifostine for mucosal and hematopoietic protection, as well as a combination of carboplatin, taxol, radiotherapy in patients with head and neck cancer (NCT00270790, www.clinicaltrial.gov), showed that amifostine significantly reduced the side effects of the treatment. To date, the FDA has approved amifostine for limited clinical indications but not for nonclinical uses. Moreover, despite the progress made to improve the effectiveness of amifostine as a radioprotector, none of the strategies have resolved the issue of its toxicity/side effects (for instance, nausea and vomiting can be frequent and severe). Amifostine is a type of organic thiophosphate prodrug. It is hydrolyzed in vivo using alkaline phosphatase into the active cytoprotective thiol metabolite, WR-1065. Scavenging of free radicals, hydrogen donation, the repair of damaged DNA, and Warburg-type and HIF1α-dependent effects may underly its radioprotective role [46]. Amifostine also shows selectivity for normal versus cancer cells, probably because of the elevated pH and upper alkaline phosphatase activity in normal tissues [47]. Different combinations that include amifostine have been tested. Synergism or clear additive effects have been observed with, e.g., zinc aspartate [48], some polysaccharides [49], 2-mercaptopropionylglycine [50], sodium selenite [51], metformin [52], peptidoglycan [53], and γ-tocotrienol [54]. Experimentally, other thiol-containing molecules, such as cysteine, *N*-acetylcysteine, cysteamine, and cystamine, have also been found to be promising in amending a large range of side effects produced by radiotherapy [55].

#### 3.1.2. Polyphenolic Phytochemicals

Natural polyphenols are plant-derived organic chemicals structurally characterized by the presence of two or more phenol units [56]. Their primary role is to protect plants from any type of stress, such as ultraviolet (UV) radiation, pathogens aggression, low fertility in the soil, too elevated or too low temperatures, severe drought, and grazing pressure [57,58]. Different natural polyphenols seem to exert radioprotection through antioxidant and free-radical-scavenging activities [59]. For instance, survival increased significantly in mice pretreated with genistein (25–400 mg/kg, subcutaneous), 24 h before being subjected to γ irradiation (9.5 Gy) [60]. Green tea polyphenols have been shown to protect the gastrointestinal tract against γ rays [61]. Survival increased by 50% in mice pretreated with 3,5,4′-tri-*O*-acetylresveratrol (10 mg/kg, i.p.), 10 min before being subjected to γ irradiation (9.7 Gy) [62]. The administration of naringenin (50 mg/kg, oral) over the 7 days before exposure to γ radiation safeguarded mice in face of chromosomal and DNA damage [63]. However, none of the many polyphenols tested to date exert truly effective radioprotection (see, e.g., [64]). Due to the fact that polyphenols have limited water solubility, a rapid metabolism under in vivo conditions and the chemical stability and solubility of polyphenols are essential [65,66]. Dietary polyphenols are quickly metabolized in tissues and the colonic microbiota [57,67]. Polyphenols follow conjugation with glucuronide, sulfate, and methyl groups (phase II metabolism); thus, just a trivial percentage of free polyphenols can be detected in the blood. Their metabolites could be found in the urine and biliary secretion, which are partly recycled by the intestinal tract. These metabolites are, in general, less effective as antioxidants than the original natural molecules [68]. Overcoming these limitations is essential if we want to exploit the properties of these molecules as radioprotectors. Structural modifications of natural molecules (e.g., in the form of salts) to increase their hydrosolubility for intravenous administration or oral formulations to increase their bioavailability (e.g., cocrystals) are two feasible options.

Eventually, topically administered polyphenols may provide strong antioxidant protection. For instance, topically administered pterostilbene (3,5-dimethoxy-4′-hydroxystilbene) fully protects against chronic UVB (180 mJ/cm^2^ × 3 doses/week × 7 months)-induced carcinogenesis in SKH-1 hairless mice [69]. The underlying mechanism involves a pterostilbene-induced increase of the physiological antioxidant mechanisms of the skin [69]. Other polyphenols that were adopted with the same protocol (e.g., resveratrol, curcumin, epigallocatechin gallate (EGCG), epicatechin, apigenin, genistein, ellagic acid, and lutein) did not yield photoprotection [70].

#### 3.1.3. Nonpolyphenolic Phytochemicals

Many other phytochemicals of a nonpolyphenolic chemical nature have also been tested as potential radioprotectors [64]. However, on the basis of experimental evidence, very few of these phytochemicals show relevant in vivo radioprotective effects. For instance, caffeine is a methyl xanthine derivative that shows antioxidant and anti-inflammatory characteristics [71]. In cancer patients, an inverse correlation was observed between caffeine ingestion and decreased severe late toxicity after radiation of the pelvis [72]. Whether the mechanisms of radioprotection may be common to methylxanthines as a drug class remains an open question. Caffeine also protected mice after having been exposed to lethal doses of γ irradiation [73]. Sesamol (3,4-methylenedioxyphenol), a component found in sesame seeds and oil, has been demonstrated to safeguard the hematopoietic and gastrointestinal systems in mice receiving toxic ionizing radiation [74]. Administration of 3,3’-diindolylmethane (DIM), a small molecule formed by acid hydrolysis in the stomach of indole-3-carbinol (a component of different cruciferous vegetables), protected mice against lethal doses of total body irradiation up to 13 Gy in a multidose schedule. However, DIM did not protect human breast cancer xenograft tumors against radiation [75]. These results are promising, but the efficacy of DIM as a potential radioprotector still requires further research. In this sense, special caution is given to its common side effects (i.e., headache, nausea, vomiting, and diarrhea).

#### 3.1.4. Vitamins

Vitamin A and β-carotene have shown radioprotective properties (reduction of mortality and morbidity) in mice exposed to partial or total-body irradiation (TBI) [76]. Moreover, dietary vitamin A (150,000 IU/kg of diet) was demonstrated to offer protection for the mice in face of localized radiation exposure to the intestine (13 Gy, TBI) and to the esophagus (29 Gy) [77].

Oral pretreatment with ascorbic acid prevented gastrointestinal syndrome in mice after being exposed to a lethal dose of radiation [78]. Immediately after exposure, administrating 3 g/kg of ascorbic acid significantly improved mouse survival following TBI at 7–8 Gy. Treatments longer than 36 h, however, did not show any effectiveness [79].

Subcutaneous (SC) injection of Vitamin E (α-tocopherol, 1 h prior to or during 15 min post irradiation (^60^Co, 0.2 Gy/min), significantly increased the 30-day post-irradiation survival in CD2F mice [80]. However, vitamin E comprises a variety of eight different isoforms that could be divided into two groups: four of them are saturated analogues (α, β, γ and δ) under the name of tocopherols, and the other four are unsaturated analogues called tocotrienols [81], all of which are collectively grouped together as tocols. Singh et al. showed tocols administered SC, 1 h prior to or during 15 min post irradiation (0.2 Gy/min), improved the 30 day survival of the mice in a significant manner [81]. Mouse protection was improved with 400 mg/kg of α-tocopherol administered SC 24 h prior to ^60^Co γ-irradiation (0.6 Gy/min) [82]. In vivo radioprotection has also been shown using different α-tocopherol derivatives, such as α-tocopherol succinate and α-tocopherol monoglucoside [81]. On the other hand, δ-tocotrienol, given as a single SC injection prior or following ^60^Co γ-irradiation, was shown to have significant protective effects on the mice in an experiment of 30-day survival [83]. Conversely, administration of γ-tocotrienol, a powerful β-Hydroxy β-methylglutaryl-CoA (HMG-CoA) reductase inhibitor, in 100 and 200 mg/kg doses, 24 h prior to ^60^Co γ-irradiation, showed a significant protective effect on the mice facing radiation doses as high as 11.5 Gy [84]. Furthermore, the combination of γ-tocotrienol and pentoxifylline (a xanthine derivative administered to patients suffering from peripheral artery disease, as a drug to attenuate muscle pain) showed a positive effect on the survival of the mice facing ^60^Co γ-irradiation in a significant manner (doses as high as 12 Gy) compared to any of the compounds alone [85]. However, despite the fact that, in different studies, tocotrienols showed a higher radioprotective effect compared to tocopherols, their low bioavailability is an important limiting factor when it comes to their clinical use as radioprotectants [86]. More importantly, the latest research indicates a high-dose injection of α-tocopherol (75 mg/kg) may cause negative effects in non-human primates, eventually leading to their death [87]. These results caution the use of vitamin E and its derivatives since, at elevated (potentially radioprotective) concentrations, vitamin E may have toxic effects on humans. Nevertheless, a pentoxifylline and vitamin E combination has shown efficacy for superficial radiation-induced fibrosis in a phase II clinical trial [88], a fact that warrants further investigation.

#### 3.1.5. Oligoelements

Many endogenous defense enzymes contain trace elements, e.g., superoxide dismutase (SOD) and metalloproteins. These enzymes help to remove radiation-induced ROS. The main oligoelements demonstrated to have protective effects in face of radiation-induced DNA damage are zinc, copper, manganese, and selenium [89].

Se (as a cofactor for many enzymes, such as glutathione peroxidase, thioredoxin reductase, and ribonucleotide reductase) and its derivatives have shown particularly strong radioprotective effects in mice. An intraperitoneal injection of sodium selenite and selenomethionine prior to (24 h and 1 h) or immediately following (>15 min) the radiation exposure improved irradiated mice’s survival rates (^60^Co, 0.2 Gy/min). In equitoxic doses (one-fourth LD10; sodium selenite = 0.8 mg/kg, selenomethionine = 4.0 mg/kg), both drugs improved the 30 day survival of mice irradiated at 9 Gy. On the other hand, survival following 10 Gy exposure was significantly dropped after selenomethionine treatment only [90]. Nevertheless, it is still an open question whether the results may indicate an advantage against the mortality sustained during nuclear emergencies [91].

A derivative of selenium, 3,3-di-selenopropionic acid (at an IP dose of 2 mg/kg for 5 days prior to whole-body exposure to γ radiation) also showed radioprotection in mice by decreasing DNA damage and apoptosis [92]. Another recent formulation, poly(vinylpyrollidone)- and selenocysteine-modified Bi_2_Se_3_ nanoparticles, improved the radiotherapy efficacy against tumors and exerted radioprotection in normal tissues [93].

Some clinical evidence indicates that oligoelement supplementation may act as an effective radioprotector in patients during radiotherapy. For instance, in a randomized clinical study, patients treated with zinc sulfate suffered a lower degree of mucositis compared to the placebo group [94]. Orally-administered zinc l-carnosine also reduced oral mucositis and xerostomia in head and neck cancer patients compared to the controls [95]. Cancer patients who took selenium selenite orally showed a lower incidence of diarrhea compared to the placebo group [96]. Selenomethionine also reduces mucositis in patients with advanced head and neck cancer who are receiving cisplatin and radiation therapy (NCT01682031, www.clinicaltrials.gov).

#### 3.1.6. Superoxide Dismutase

SOD, a unique type of enzyme whose principal role is superoxide radical dismutation, has shown radioprotective properties. Preliminary research showed that a bovine SOD IV injection to mice promoted erythrocyte, reticulocyte, and white blood cell recovery after an X irradiation-induced loss [97]. This observation is in agreement with later experiments that correlated the radioprotection in hematopoietic stem cells with the IL-1-induced increase in SOD2 activity [98]. As reported by Yan et al. [99], when a single dose of 35 Gy was injected to mice, the SC injection of AAV2-Mn-SOD-hrGFP (a recombinant adeno-associated virus vector expressing SOD2 tagged with humanized recombinant green fluorescent protein) showed significant mitigation and enhanced the speed of healing process, in 2 weeks after radiation in comparison with the control group. Since that primary observation, different reports further supported this strategy (see, e.g., [100]). For instance, studies on the effect of SOD2 gene-modified mesenchymal stem cells (MSCs) on improving recovery of radiation-induced lung injury have shown such efficacy [101]. Nevertheless, the need for inducing in vivo gene expression indicates technological limitations.

A third-generation, cationic, lipophilic, Mn, and porphyrin-based mimetic of the human SOD2 is represented by BMX-001 (BioMimetix, Englewood, CO, USA). After administration, BMX-001 is internalized by the cells and imitates the SOD2 activity by scavenging ROS. BMX-001 can potentially interact with numerous redox-sensitive pathways, for example, those involving NF-κB and Nrf2, thus having an impact on cellular transcriptional activity [102]. BMX-001 has been shown to protect the brain’s white matter against ionizing radiation while acting as a tumor radiosensitizer [103]. More specifically, the administration of BMX-001 for one week before cranial irradiation and continuing for one week afterward supported the long-term survival of newborn neurons in the hippocampal dentate gyrus [104]. BMX-001 has been assayed as a radioprotector in different clinical trials (www.clinicaltrials.gov), e.g., NCT03386500 (patients with recently diagnosed anal cancer), NCT03608020 (cancer patients with multiple brain metastases), NCT02990468 (head and neck cancer), and NCT02655601 (high-grade glioma treated with concurrent radiation therapy and temozolomide).

#### 3.1.7. Ex-Rad

A synthetic chlorobenzylsulfone derivative named recilisib sodium (Ex-Rad) is being studied as a protector against radiation by Onconova Therapeutics (Newtown, PA, USA) and the US Department of Defense. In this regard, 30 day survival studies using C3H/HeN male mice demonstrated a 88% survival rate in cases that 500 mg/kg of Ex-Rad was administered SC 24 h and 15 min prior to γ irradiation with 8.0 Gy [105]. Further experiments concluded that both SC and oral Ex-Rad provided a significant survival benefit against radiation toxicity and that Ex-Rad-mediated hematopoietic and gastrointestinal protection has a considerable impact on enhancing the mice’s survival [106]. Mechanistic studies suggest that Ex-Rad shows its properties via upregulating the PI3-kinase/AKT pathway radiation-exposed cells [107], a fact that is cause for concern, as this pathway is associated with all three major radiation resistance mechanisms in cancer cells: intrinsic radiosensitivity, tumor cell proliferation, and hypoxia [108]. Nevertheless, recent experiments have shown that salidroside, a glucoside of tyrosol and an ROS scavenger, improves the radioprotective effect of Ex-Rad through a p53-dependent apoptotic pathway [109].

#### 3.1.8. Nitroxides

Nitroxide free radicals may protect cells exposed to oxidative stress, such as superoxide and H_2_O_2_ [110]. Nitroxides’ ability of catalyzing the dismutation of superoxide radicals (SOD-like activity), preventing Fenton and Haber–Weiss reactions through the oxidation of transitioning metal ions to a more elevated oxidative state, and giving catalase-like activity to heme proteins, gives them an antioxidant effect for inhibiting lipid peroxidation. Preclinical studies have demonstrated that the a nitroxide’s oxidized form could be an in vivo radioprotector according to models subjected to lethal TBI [111]. The lead compound from this class is tempol (4-hydroxy-2,2,6,6-tetramethylpiperidine-1-oxyl), which has been found to prevent radiation-induced micronuclei formation (in peripheral blood leucocytes) and chromosomal aberrations (in bone marrow cells) [112], as well as irradiation-induced mucositis, in experimental animals subjected to TBI [113]. The toxic dose of tempol is approximately 1–2 mmol/kg (IP or IV) in different animal species (see, e.g., [114]). This toxicity represents a limitation and questions if a human equivalent dose could be tolerated in humans and show radioprotective efficacy. In the clinical setting, a phase II trial to assess the ability of tempol (600 mg/day) to prevent and/or reduce the toxicity associated with cisplatin and radiation treatment is being performed in head and neck cancer patients (NCT03480971, www.clinicaltrials.gov), but this study remains ongoing.

#### 3.1.9. Hormones and Hormone Analogues

Melatonin, *N*-acetyl-5-methoxytryptamine, protects DNA, lipids, and proteins from free-radical damage. The effect of exogenous melatonin in reducing the oxidative stress and inflammation resulting from ionizing radiation has been demonstrated in both in vitro and in vivo studies of a variety of species [115]. Early experiments showed that lymphocytes collected from human subjects after orally receiving a dose of 300 mg of melatonin present a clear decrease in chromosomal aberrations after irradiation [116]. Pretreatment with melatonin (250 mg/kg) also protected mice subjected to lethal irradiation doses (8.15 Gy) (85% survival versus 45% in the controls) [117]. Nevertheless, its short half-life in vivo (<1 h) and the need for high doses to achieve radioprotective effects create challenges and limits in practice. Consequently, very little has been done regarding clinical trials. Nevertheless, it has been reported that the administration of melatonin with conventional treatment reduced severe oral mucositis development in neck cancer patients undergoing radiotherapy [118].

Indraline, an α-adrenomimetic (α-1(B)-adrenoagonist), showed clear radioprotective effects for the skin, certain organs, and cellular DNA exposed to a radioactive source in different animal species [119]. The potential radioprotective effects of indraline include, e.g., an increase in DNA and protein biosynthesis and an increase in ribonucleotide reductase [120]. Indraline-treated animals survived better (at least in the short term) after external irradiation, even at normally fatal doses, especially if certain vital organs (including the liver) were also protected [121]. Importantly, indraline has been shown to be radioprotective in irradiated monkeys. Experimentally, juvenile rhesus monkeys were exposed to total-body γ irradiation from ^60^Co at a dose of 6.8 Gy (100% lethality over 30 days). Five minutes before the exposure, an intramuscular dose of indraline (40–120 mg/kg) was administered. Five out of six monkeys were protected by the 120 mg/kg does of indraline (in comparison with the 10 animals that died in the radiation control group) [122]. Although the lack of clinical trials is a very limiting factor, it is encouraging that, in experiments on rats, indraline (50 mg/kg) and mexamine (12 mg/kg) injected IP almost completely eliminated animal mortality from the intestinal syndrome of acute radiation sickness [123].

#### 3.1.10. Antibiotics

Primary experiments performed in the 1960s reported that antibiotic treatment and a single transfusion of allogeneic platelets can significantly reduce the mortality of monkeys exposed to a potentially lethal dose of total-body X irradiation [124]. Later, experiments in the 1970s demonstrated that, 2 weeks before supralethal TBI, administration of streptomycin, kanamycin, neomycin, or gentamycin in the drinking water (4 mg/mL) of specific pathogen-free C57×Af mice, very significantly prolonged the mice’s mean survival times (8.2–8.9 days vs. 6.9 for the controls) up to values far beyond those registered for germ-free mice (7.3 days) [125].

More recently, high-throughput screening identified two types of antibiotics, tetracyclines and fluoroquinolones, as potential radioprotectors and mitigators of the hematopoietic system [126]. Murine hematopoietic stem/progenitor cell populations were protected from radiation damage by tetracycline; 87.5% of mice survived when receiving it prior to and 35% when given 24 h following lethal TBI. Of note, the radiosensitivity of, for example, Lewis lung cancer cells was not altered by tetracycline [126]. Tetracycline hydrochloride demonstrates free-radical-scavenging activity, protects DNA, and increases survival (C57BL/6 male mice) by 20% at a lethal radiation dose of 9 Gy [127]. Nevertheless, antibiotics as radioprotectors have not yet been tested in clinical trials. Moreover, their side effects at potentially radioprotective doses may represent a limiting factor for their efficacy.

#### 3.1.11. Adenosine Receptor Agonists

The protective impact of elevating extracellular adenosine on early cellular damage in 1 Gy irradiated mice was first reported in the early 1990s [128]. The authors suggested that AMP, an adenosine prodrug, and dipyridamole, a drug that inhibits adenosine uptake by cells, used concurrently, would increase extracellular adenosine and then activate cell surface adenosine receptors. There could be radioprotective effects resulting from the systemic (vasodilation, hypotension, hypoxia) and cellular (elevation of cAMP in sensitive cells) consequences of adenosine receptor activation [128]. Further experiments showed that the combined preirradiation administration of AMP and dipyridamole had a positive effect on hematopoiesis and improved survival [129,130]. Later, it was discovered that a specific adenosine A3 receptor agonist (*N*^6^-(3-iodobenzyl)adenosine-5′-*N*-methyluronamide (IB-MECA)) acts as a homeostatic regulator of bone marrow hematopoiesis [131]. Higher hematopoiesis activation was observed by combining the A3 receptor agonist with the granulocyte colony-stimulating factor [132] or meloxicam [133]. This last combination promoted survival in mice subjected to lethal radiation doses [134]. Protection of the spleen of γ-ray-irradiated mice against the immunosuppressive and oxidative effects of radiation by AMP has also been recently reported [135]. IB-MECA is being developed for the treatment of autoimmune anti-inflammatory diseases. In phase II trials, its action in rheumatoid arthritis and psoriasis compared favorably to that of existing treatments for those conditions, but it did not display serious adverse effects [136]. This indicates good potential for its testing as a radioprotector.

#### 3.1.12. DNA-Binding Molecules

There are two ways in which drugs bind to DNA: covalently or noncovalently. Most of the noncovalently bound drugs are from two major types: minor groove binders and intercalators [137]. All these drugs display different degrees of cytotoxicity and are mainly used in cancer therapy, although a few (paradoxically) promote cell survival and radioprotection [138].

Following some initial observations showing that the minor groove ligand Hoechst 33342 has radioprotective properties [139,140], some analogues were designed to improve radioprotective activity. Proamine and, particularly, methylproamine [141] showed greater efficacy compared to Hoechst 33342. Netropsin, which is also a minor groove-binding ligand, showed radioprotective properties linked to a mechanism favoring DNA stability [142]. Nevertheless, these drugs, due to their molecular interactions with DNA, may also exert potential promutagenic effects, which could lead to cancer development [143]. Pentamidine, a minor groove binder and antimicrobial drug also used to prevent and treat pneumocystis pneumonia in patients with poor immune defenses via the inhibition of histone H2Aacetylation, represses DNA damage responses and shows no mutagenic effects in DNA [144]. Thus, minor groove binding does not necessarily lead to mutagenesis, and further studies are required to ascertain the role of these molecules as safe radioprotectors.

### 3.2. Radiomitigators

#### 3.2.1. Glutamine

l-Gln is an amino acid naturally occurring in the human body. It is the most abundant nonessential amino acid and provides the main metabolic fuel for small intestine enterocytes, lymphocytes, macrophages, and fibroblasts [145].

Supplementation of l-Gln during cancer treatment reduces the incidence of gastrointestinal, neurological, and possibly cardiac complications and potentially minimizes toxicity resulting from or related to chemotherapy [146,147]. In rats, the provision of a diet enriched with l-Gln during TBI (10 Gy, a dose that causes a 50% mortality rate in a few days) was shown to protect the small-bowel mucosa. This protection happens by providing support for crypt cell proliferation, thus making the healing process of the irradiated bowel much faster; it was shown to reach 100% survival among the irradiated subjects [146]. However, a meta-analysis of 13 randomized controlled trials concluded that l-Gln failed to improve the severity and symptoms (in terms of tenesmus, abdominal cramping, and blood in bowel movements) in patients with radiation enteritis [148]. Nevertheless, the combined administration of diets rich in l-Gln- and l-Arg was shown to have protective effects on the gut mucosa during the postirradiation state. In all of the groups using l-Gln and l-Arg, the intestinal villus count and villus height were significantly higher than those of the control group [149]. Furthermore, the combined supplementation of β-hydroxy-β-methylbutyrate, l-Gln, and l-Arg further improved acute intestinal damage caused by radiation [150]. Moreover, l-Gln has also been effective in decreasing the incidence and severity of oral mucositis caused by the chemo-radiotherapy, and dysphagia in patients with locally advanced oropharynx and larynx carcinoma [151]. In addition, l-Gln supplementation has the potential of attenuating the harshness of diarrhea related to chemotherapy, neuropathy resulting from paclitaxel, and hepatic veno-occlusive disease because of high-dose chemotherapy and stem-cell transplantation [147]. These potential benefits of l-Gln, alone or in combination, need further clinical testing.

#### 3.2.2. Probiotics

A probiotic is a preparation with large amounts of viable and defined microorganisms, enough to alter the host’s microflora [152]. As shown by preliminary observations in preclinical models, probiotics may help mitigate radiation-induced gastrointestinal damage [153,154]. Some clinical studies examined the effects of different probiotics, showing a significant reduction in diarrheal symptoms [155]. As concluded in 2014, some limitations in these studies, including the number of patients, did not allow their results to support the use of any specific probiotic as a radiomitigator of radiation-induced gastrointestinal syndrome. Nevertheless, a recent review with a meta-analysis of randomized controlled trials on the incidence of diarrhea in cancer patients receiving radiation therapy concluded that probiotics are indeed beneficial against radiation-induced diarrhea [156]. In this regard, choosing the right probiotic can be crucial, because preparations vary significantly. Probiotics may lack sufficient bacteria, contain dead bacteria, or simply offer the wrong species of bacteria.

Following doses capable of producing death from gastrointestinal syndrome, intestinal crypt cells loss, principally leucine-rich repeat-containing G-protein-coupled receptor 5 stem cells, could be detected [157]. Recent advances demonstrated that second-generation probiotics producing IL-22 (specifically, *Lactobacillus reuteri* or *Escherichia coli* harboring the transgene for IL-22) increased the survival of mice after TBI [158]. C57BL/6 mice receiving IL-22 probiotics at 24 h following 9.25 Gy TBI presented green fluorescent protein-positive bacteria in the intestine, increased the number of leucine-rich repeat-containing G-protein-coupled receptor 5-expressing intestinal stem cells to twice their original amount, and had increased 30 day survival [158].

#### 3.2.3. Angiotensin-Converting Enzyme Inhibitors and Angiotensin II Receptor Blockers

Angiotensin-converting enzyme (ACE) is a zinc metalloenzyme that is central in the renin–angiotensin system. ACE inhibitors constrain ACE and reduce the formation of angiotensin II from angiotensin I (AT1). ACE inhibitors are extensively utilized for the treatment of hypertension, heart failure, diabetic nephropathy, and type 2 diabetes mellitus. It has also been reported that ACE inhibitors, e.g., perindopril, protect against irradiation-induced death [159]. Pretreatment using perindopril enhanced bone marrow cellularity and increased the number of hematopoietic granulocyte macrophage progenitor colony-forming units, erythroid burst-forming units, and megakaryocyte colony-forming units from days 7 to 28 following irradiation [159], thus suggesting that ACE inhibitors could be candidates to mitigate the hematopoietic toxicity of irradiation. Clinical trials have shown the potential of ACE inhibitors to reduce radiation-induced pneumonitis and fibrosis (enalapril, NCT01754909, www.clinicaltrials.gov).

Further experiments using the ACE inhibitor captopril concluded that, through modulating the hematopoietic progenitor cell cycle, it promoted hematopoietic recovery process after radiation. However, the timing of captopril treatment relative to radiation exposure affects the viability and repopulation capacity of spared hematopoietic stem cells in different ways. Therefore, it may result in radiation protection or radiation sensitization [160]. In this sense, more recent experiments using C57BL/6 mice exposed to an LD_50–80/30_ of ^60^Co TBI (7.75–7.9 Gy) showed that administration of low-dose captopril, started as late as 48 h post-TBI and sustained for 14 days, enhanced overall survival rates in a significant manner, comparable to high-dose rapid administration [161]. Captopril has also been studied as a radiation mitigator for the lungs, kidneys, brain, and skin [162,163].

Angiotensin receptor blockers (ARBs) impede the function of the angiotensin AT1 receptor and lower the actions of angiotensin II. For instance, DAA-I (des-aspartate-angiotensin I), an orally active angiotensin peptide, works as an agonist on the AT1 receptor and brings about responses that counter those of angiotensin II. Therefore, DAA-I has been studied because of its potential radioprotection in γ-irradiated mice. DAA-I functions through a unique mechanism of action on the angiotensin AT1 receptor to particularly release prostaglandin E2, which acts and a mediator for radioprotection in γ-irradiated mice [164]. ARBs show the most promise in areas with high levels of ACE (lung tissue) or at the sites of angiotensin II action (kidney) [165]. A recent phase III clinical trial assayed the effects of an AIIRB (losartan), steroids, and radiotherapy in glioblastoma (NCT01805453, www.clinicaltrials.gov). This study concluded that losartan was well tolerated, although it did not reduce the steroid requirements in newly diagnosed glioblastoma multiforme patients treated with concomitant radiotherapy and temozolomide.

#### 3.2.4. Statins

Statins are inhibitors of hydroxymethylglutaryl-coenzyme A reductase (HMGCR). It is an enzyme that rate-controls the mevalonate pathway, which produces isoprenoids such as cholesterol. They are effective in the therapy of hypercholesterolemia and, thus, are a common medication for people at high risk of cardiovascular disease, as well as in those who have already have developed a cardiovascular disease [166]. Moreover, it has been demonstrated that statins decrease the messenger RNA (mRNA) expression of proinflammatory and profibrotic cytokines stimulated by ionizing radiation in vitro and mitigate ionizing radiation-induced inflammation and fibrosis in vivo. Statins may reduce radiation-induced endothelial activation and the subsequent inflammatory and thrombotic responses [167]. Moreover, statins increase the rapid repair of DNA double-strand breaks and, at the same time, alleviate the DNA damage response induced by ionizing radiation [168,169,170]. Statins also act on Rho/Rho-associated protein kinases after radiation therapy, decreasing fibrosis via connective tissue growth factors [171]. On the basis of these facts, it was proposed that statins could be considered in therapeutic strategies for the management of patients treated with radiation therapy. Interestingly, a recent study showed that patients with hyperlipidemia and radiotherapy who used statins or metformin after prostate cancer diagnosis had longer average survival times [172]. Nevertheless, due to restrictions of the database used, information including patient lifestyle, family history, patient demographics, cancer stage, and clinical laboratory information was not available and could not be evaluated. Therefore, more studies are necessary to further assess the potential benefit of this combination.

#### 3.2.5. Somatostatin Analogues

One of the main issues in gastrointestinal toxicity is the severity of mucosal damage. Apoptosis and clonogenic cell death, depending on the severity of injury, may result in cytokine-activated inflammatory reactions and peripheral immune cell activation [173]. This radiation vulnerability could be to a certain extent because of the presence of the pancreas. Digestive pancreatic enzymes can increase inflammation and the damage to subepithelial tissues [174]. Therefore, reducing intraluminal proteolytic activity may help attenuate intestinal radiation toxicity.

Somatostatin and its analogues can inhibit exocrine pancreatic secretions [175]. Thus, some compounds have been tested for their efficacy. For instance, the synthetic somatostatin analogue, octreotide (an octapeptide), seems to ameliorate intestinal radiation injury (both acute and delayed) [176,177] and, particularly, gastrointestinal diarrhea [178]. A randomized controlled trial in which octreotide was tested to prevent diarrhea in patients who were undergoing radiation therapy to the pelvis (NCT00033605, www.clinicaltrials.gov) concluded that octeotride is more effective than conventional therapy with diphenoxylate and atropine in controlling acute radiation-induced diarrhea. Nevertheless, some common side effects may limit the efficacy of octreotide. Its dose-limiting toxicity was found to be mainly allergic (nausea, rash, and lightheadedness) and endocrine (hypoglycemia) [179].

Another somatostatin analogue SOM230 (pasireotide), under preclinical evaluation, ameliorated intestinal mucosa injury and increased mouse survival after TBI by inhibiting exocrine pancreatic secretion [180]. SOM230, with a 40-fold improved affinity to somatostatin receptor 5 than other somatostatin analogues, not only proved to be beneficial when started prior to radiation exposure, but it also attenuated radiation toxicity in cases where it was started up to 48 h following exposure [181].

#### 3.2.6. Immunomodulators

There is evidence indicating that a few bacterial species are radioresistant [182]. Glucans, β-glucan in particular, are bacterial wall constituents [183]. The hematopoiesis-stimulating effects of β-glucan could be adopted in treating hematopoietic ARS in mice. Importantly, the combined use of amifostine or cysteamine and β-glucan exerts additive radioprotective and radiomitigating effects [184,185]. Positive effects have also been observed by combining β-glucan and selenium [186], a cyclooxygenase (COX) inhibitor [187], or the granulocyte colony-stimulating factor [188]. Post-irradiation hematopoiesis and survival have been further confirmed [189], and the use of β-glucan as a therapy in nuclear emergencies has been proposed [190]. There have been no reported side effects from taking beta-glucans orally, but they have very limited bioavailability. However, when used via injection, beta-glucans can cause chills, fever, headache, back and joint pain, nausea, vomiting, diarrhea, dizziness, high or low blood pressure, flushing, rashes, tiredness, a decrease in the number of white blood cells, and an increase in the production of urine [191].

Peptidoglycan (a bacterial wall polymer), when injected after irradiation in mice, can increase survival and amend intestinal and bone marrow damage [53]. Moreover, as indicated above, synergism was also observed when amifostine was given as a radioprotector alongside peptidoglycan as a radiomitigator [53].

A natural adrenocortical hormone, 5-androstenediol, stimulates hematopoiesis and the innate immune system in γ-irradiated mice [192] and monkeys [193]. Hematopoiesis stimulation appears to be related to a 5-androstenediol-induced increase in the granulocyte colony-stimulating factor [194] and in different cytokines and thrombopoietin in the bone marrow [195]. These and other observations highlight 5-androstenediol as a likely candidate for the treatment of ARS. Recently, experiments in mice have shown that 5-androstenediol can prevent radiation injury by encouraging NF-κB signaling and inhibiting inflammasome-mediated pyroptosis [196]. Importantly, this radioprotective effect is accompanied by low toxicity.

CBLB502, a recombinant protein acting as an agonist of Toll-like receptor 5 (TLR5, an innate immunity receptor), activates TLR5 and triggers NF-κB signaling. This mobilizes an innate immune response that drives the expression of numerous genes, i.e., apoptosis inhibitors, ROS scavengers, and different regenerative cytokines [197]. CBLB613, a lipopeptide obtained from *Mycoplasma arginini* and a TLR2/6 agonist, has also demonstrated substantial radioprotective capacity in CD2F1 mice against hematopoietic syndrome [198].

Treating mice with sepsis using antibiotics and synthetic trehalose dicornomycolate (a glycolipid found in the cell wall of *Mycobacterium tuberculosis*) after irradiation and trauma has been tested with success [199].

Despite all these potentially protective effects, immunomodulatory drugs may have oral and systemic adverse effect [200], and there are currently no properly powered randomized controlled trials published on radiomitigation.

#### 3.2.7. Cytokines

Early experiments in the 1980s showed that IP administration of IL-1 exerted strong protection in C57BL/6 mice after a LD_50/30_ radiation dose [201]. Further experiments showed that antibodies against interleukins and TNF-α decreased CD2F1 mice’s survival after TBI because of myeloid suppression [202]. Following this research, different cytokines (i.e., IL-1, IL-12, TNF-α, basic fibroblast growth factor, and G-CSF) have shown protective effects in mice when given prior to irradiation [203]. TGF-β3, which is involved in regulating pulmonary fibrosis, has shown efficacy in preventing radiation-induced damage. IP administration of TGF-β3 decreased pulmonary fibrosis after thoracic irradiation (20 Gy) in mice [204]. Nevertheless, different toxicity- and pharmacokinetic-related issues limit their potential as radiomitigators, at least as a monotherapy.

G-CSF and GM-CSF have been demonstrated to improve neutropenia induced by cancer therapy, and recent advances have yielded filgrastism (a G-CSF produced by recombinant DNA technology), pegfilgrastim (a PEGylated form of the G-CSF analogue filgrastim), and sargramostim (a recombinant GM-CSF), which are currently under study as radiation countermeasure agents [198].

Palifermin, a truncated human recombinant keratinocyte growth factor (KGF) generated in *Escherichia coli*, can stimulate proliferation and differentiation of epithelial cells and upregulate cytoprotective mechanisms [205]. FDA initially approved palifermin to prevent oral mucositis in cancer patients who receive hematopoietic stem-cell transplants. Later, it was also shown to be effective in minimizing the severity of mucositis in head and neck cancer patients treated with chemotherapy and radiotherapy [206]. R-spondin-1 (RSPO1), a human protein required for the early development of the gonads, elevated basal cellularity, thickened the mucosa, and elevated epithelial cell proliferation in the tongue. RSPO1 has also shown efficacy against chemoradiotherapy-induced mucositis, in addition to promoting the proliferation gastrointestinal epithelial cell in mice [207].

The radiation necrosis of normal CNS tissue is one of brain irradiation’s main risk factors. The same is true for radiation-induced necrosis results because of the surge in capillary permeability resulting from cytokine release, causing extracellular edema. Thus, it has been suggested that blocking the VEGF could reduce radiation-induced necrosis by reducing vascular permeability [208]. In the first attempt, it was reported that bevacizumab (an anti-VEGF antibody) reduced brain necrosis in a patient subjected to cranial irradiation [209]. Further experiments concluded that, for the management of edema associated with radiation necrosis, a low-dose bevacizumab could be a potent measure [210,211].

Although different cytokines have been assayed in combination with other drugs in trials where radiation was also administered, there are still no reliable reports that specifically address the effects of any of these molecules as general radiomitigators.

#### 3.2.8. Prostaglandins and Nonsteroidal Anti-Inflammatory Drugs (NSAIDs)

Prostaglandins (PGs), as part of the eicosanoid family, are signaling molecules produced by arachidonic acid (or any other similar polyunsaturated fatty acids) oxidation (enzymatic and nonenzymatic). Early studies in the 1980s showed that PGE2 (alone or in combination with amifostine) and misoprostol (a synthetic PGE1 analogue) can radioprotect the gastrointestinal tract in mice [212,213]. However, the potential in vivo inhibition of myelopoiesis induced by PGE2 [214] precludes its use as a useful radioprotector or radiomitigator.

Nonsteroidal anti-inflammatory drugs (NSAIDs) show anti-cancer effects by producing cell-cycle arrest, thereby shifting cells toward a quiescent state (gap 0 (G0)/G1). In normal tissue radioprotection, the same mechanism of action was detected, which suggests that there is possibility of NSAIDs shielding normal tissues in face of radiation injury [215]. There are two main categories of NSAIDs available: nonselective and COX-2-selective. Nonselective COX inhibitors (which interfere with the synthesis of PGs from arachidonic acid) have shown hematopoiesis-stimulating effects in sublethally irradiated experimental animals [216]. However, therapy with nonselective COX inhibitors associated with severe gastrointestinal side effects compromised the survival of mice subjected to lethal doses of ionizing radiation [217]. Nevertheless, meloxicam (a selective COX2 inhibitor) has been shown to trigger hematopoiesis and survival in irradiated mice [218]. More recently, sequential administration of PGE2 and meloxicam was shown to increase hematopoiesis and survival in irradiated mice [219].

Early studies demonstrated that long-term treatment with acetylsalicylic acid (which inhibits COX-1 irreversibly and changes the COX-2’s enzymatic activity) is effective in reducing renal functional impairment after irradiation [220]. The lung is an organ with certain irradiation sensitivity, and acetylsalicylic acid has been shown to reduce radiation-induced oxidative damage in lung tissue [221]. Furthermore, noteworthy structural chromosomal aberrations happened in mice exposed to 2 Gy γ radiation. Acetylsalicylic acid at 0.5 mg/kg dose (IP), injected into Swiss Albino male mice 72 h prior to γ radiation of 2 or 4 Gy doses, reduced chromosomal aberrations [222]. The low toxicity of acetylsalicylic acid and other NSAIDs favors their testing, although their use in combination with other drugs will likely render better effects.

#### 3.2.9. BIO 300

BIO 300 (Humanetics Pharmaceuticals, Edina, MN, USA), a formulation containing synthetic genistein nanoparticles, has been developed to alleviate and treat radiation-induced pneumonitis or pulmonary fibrosis. The main active ingredient in BIO 300 is genistein, which lowers collagen deposition and protects against delayed lung sequelae biomarkers after total-body or organ-specific irradiation in rodents [223,224]. The administration of BIO 300 (200 mg/kg, s.c.) has been demonstrated to decrease the formation of collagen-rich lesions in the irradiated lungs of animals surviving the hematopoietic subsyndrome of ARS [223]. Furthermore, oral administration of a dose of 400 mg/kg BIO 300 for 4–6 weeks (started 24 h after whole-thorax lung irradiation) reduced the morbidity associated with irradiation [225]. More recently, it has also been shown that BIO 300 alleviates erectile dysfunction resulting from radiation and increases human prostate cancer xenograft sensitivity to radiation therapy [226]. The results of a phase I clinical trial (NCT00504335, www.clinicaltrials.gov) aimed at assessing BIO 300 capsules’ safety and pharmacokinetics, in doses capable of producing a radioprotective or therapeutic effect in humans, are still pending. There is active research currently funded by the National Institutes of Health (USA) to develop BIO 300 as a radiation countermeasure under advanced development for acute radiation syndrome and the delayed effects of acute radiation exposure [227]. Moreover, Humanetics is also trying to develop BIO 300 for astronauts, as a radiation countermeasure and radioprotector.

#### 3.2.10. ABC294640

ABC294640 (RedHill Biopharma Ltd., Tel Aviv, Israel) is an inhibitor of sphingosine kinase-2, and, in some mouse models, it has demonstrated anticancer activity [228]. It also completed phase I trials in cancer patients with advanced solid tumors [229]. ABC294640 increases ceramide and decreases sphingosine 1-phosphate levels in tumor cells. These conditions have the potential to increase the antitumor effects of radiation. ABC also reduces radiation-induced gastrointestinal inflammation but without interfering with the antitumor activity of radiation. This provides enough support for ABC’s clinical testing in cancer patients who also go under radiotherapy [230]. ABC294640 has been explored in mice as a potential therapy for gastrointestinal ARS, but specific clinical trials are still pending. Importantly, a phase I study in patients with advanced solid tumors demonstrated that ABC294640 is well tolerated at therapeutic doses [229].

#### 3.2.11. Cell Therapy

Stem cells have the intrinsic potential to repair damaged tissues. Mesenchymal stem cells (MSCs) are a type of nonhematopoietic adult stem cell with self-renewal and multilineage differentiation potential and they show special properties, including secreting hematopoietic growth factors, reconstructing hematopoietic microenvironment, and low immunogenicity; they are cable of being effectively transduced with vectors containing therapeutic genes. MSCs have some benefits compared to other stem cells; they are much easier to isolate from the bone marrow, fat tissue, umbilical cord blood, or placenta of patients and donors and readily expanded ex vivo [231]. MSCs have shown efficacy in protecting the liver against radiation-induced injury [232], healing irradiated skin in mice [233,234], increasing survival in irradiated mice [235], and mitigating radiation-induced gastrointestinal and hematopoietic syndromes in mice [236,237]. Some key advances support this radiomitigating efficacy, e.g., MSC-derived extracellular vesicles provide long-term survival after TBI without additional hematopoietic stem cell support [238]. MSC-derived cytokines repaired radiation-induced intra-villi microvascular injury [239]. hypoxia-induced MSCs enhanced the protection of radiation-induced lung injury [240]. the transplantation of bone marrow MSCs prevented radiation-induced arterial injury [241]. MSCs attenuated radiation-induced brain injury [242]. and MSCs prevented the neurological complications of radiotherapy [243,244]. Moreover, macrophages previously exposed to exosomes from MSCs mitigated ARS by favoring hematopoietic recovery [245]. MSCs have successfully been assayed against radiation-induced pulmonary fibrosis (NCT02277145) and radiation-induced xerostomia (NCT03876197) (www.clinicaltrials.gov).

Bone marrow stromal cell transplantation has also been shown to renew the irradiated intestinal stem cell niche and alleviate radiation-induced gastrointestinal syndrome [246]. Transplanted common myeloid progenitors and megakaryocyte/erythrocyte-restricted progenitors, but not granulocyte/monocyte-restricted progenitors, protected lethally irradiated mice in a dose-dependent manner. This led to the conclusion that erythrocytes and platelets (or both of them) are critical effectors of radioprotection [247]. In CD2F1 mice exposed to different radiation doses and then transfused (IV) with mouse myeloid progenitor cells, survival improved, the level of bacterial infection decreased, and endotoxin levels in the serum were lowered (STO-MP-HFM-223 www.cellerant.com). This, therefore, indicates that myeloid progenitors have the potential to alleviate radiation injury and enhance intestinal integrity following TBI. Moreover, cryopreserved allogeneic myeloid progenitor cells significantly improved mice’s survival, in strains irradiated with lethal doses of ^60^Co γ radiation (CD2F1, 9.2 Gy) or X-ray exposure (Balb/c, 9 Gy), proven to produce acute radiation syndrome in hematopoietic tissues [248]. More recently, it has been shown that ex vivo expanded mismatched hematopoietic progenitor and stem cells have the potential of offering quick, high-level hematopoietic reconstitution, thereby attenuating ionizing radiation-induced mortality and conveying donor-specific immune tolerance in a murine hematopoietic ARS model [249].

Table 1 schematically shows the most commonly known radioprotectors and radiomitigators that have been tested in vivo and their main radiomodifying mechanisms.

### 3.3. Radionuclides and Methods to Treat Contaminated Patients

Radionuclides are atoms with an excess of nuclear energy, making them unstable. Their process of stabilization (radioactive decay) is associated with the emission of ionizing radiation. Radionuclides can enter our body either accidentally (as they are used for many purposes) or as a consequence of a nuclear explosion. In the latter case, radionuclides discharged into the atmosphere can be inhaled, ingested, or contact our skin (www.world-nuclear.org). In this context, different factors are important to determine the medical consequences of contamination by a radionuclide, including the amount entering the body, its chemical form (which influences solubility), the type of emissions and its half-life, the radiosensitivity of the organs/tissues affected, the age of the patient, and the status of the physiological mechanisms that may facilitate elimination (e.g., kidneys). Therefore, the immediate treatment must be focused on reducing the entrance of the radionuclide into the blood and its deposition in organs/tissues, accelerating its excretion, and minimizing the absorbed dose. Procedures to treat contamination with radionuclides can be grouped as described below.

#### 3.3.1. Blockers

For example, potassium iodide reduces the deposition of radioactive iodine isotopes in the thyroid gland. If administered rapidly, blockers will saturate the gland with nonradioactive iodine, thus facilitating the elimination of radioactive isotopes [297].

#### 3.3.2. Dilution

Enhanced fluid intake (e.g., water, tea, and milk) increases the excretion of, e.g., tritium and can reduce the time it stays in the body [298].

#### 3.3.3. Reducing Absorption

For instance, the absorption of strontium can be decreased by using a large amount of orally administered calcium chloride [299] or *Chlorella* [300].

#### 3.3.4. Displacement

Under this technique, a nonradioactive element will compete for the uptake sites, displacing the radioisotope from the receptor. For example, calcium gluconate competes with radiostrontium for bone deposition [301].

#### 3.3.5. Ion Exchange

For example, Prussian blue is useful to capture recycled cesium and thallium from the blood into the gut through an ion exchange mechanism. Prussian blue is a nonabsorbable resin that works as a laxative. The most effective form of Prussian blue is its colloidal soluble form [302].

#### 3.3.6. Increased Turnover

This involves increasing the natural turnover process of releasing radionuclides from organs/tissues to reduce deposition. For instance, orally administered ammonium chloride decreases the blood pH and increases the elimination of internalized radiostrontium [303]. On the other hand, sodium bicarbonate can be used to increase blood pH and facilitate the elimination of uranium [304].

#### 3.3.7. Chelators and Functional Sorbents

These compounds will bind to metals. For instance, using this technique, plutonium complexes can be more rapidly eliminated through the kidneys and the intestinal tract. Formulations of diethylene triamine pentaacetic acid (DTPA) with calcium or zinc comprise the chelator with the highest potential range of use. Other chelators in use are dimarcaprol, dimercaptosuccinic acid, and deferoxamine [305]. Alternatively, some silica-based hybrid materials (i.e., isomers of isomers of hydroxypyridinone, diphosphonic acid, acetamide phosphonic acid, glycinyl urea, and DTPA) have been proposed to remove radionuclides of plutonium, americium, uranium, and thorium from blood [306]. In the case of accidental release of radionuclides in a nuclear facility or in the environment, internal contamination (inhalation, ingestion or wound) with actinides represents a severe health risk to human beings. Guidance to assist physicians and others who may be called upon to treat workers or members of the public who may become contaminated internally with inhaled plutonium nitrate, plutonium tributyl phosphate, americium nitrate, or americium oxide can be found in [307,308].

#### 3.3.8. Surgical Excision

This process is used to remove a fixed radionuclide contaminant. However, such surgery must be carefully evaluated (risks versus benefits) and performed with the assistance of radiation protection staff [309].

#### 3.3.9. Lung Lavage

This technique will only be used for insoluble inhaled radioactive particles (e.g., plutonium) deposited in the lungs. In this case, flexible bronchoscopy should be performed for the bronchoalveolar lavage [310].

## 4. Validated and Potential Biomarkers to Assess the Harmful Effects of Ionizing Radiation

As defined by the US NIH, a biomarker is “a characteristic that is objectively measured and evaluated as an indicator of normal biological processes, pathogenic processes, or pharmacological responses to a therapeutic intervention” (www.ninds.nih.gov). Biomarkers that closely correlate with the pathophysiology of a disease are considered as promising biomarkers. The traditional radiation exposure biomarkers are based on cytogenetic assays and they continue to be used as the standard. Rapid and noninvasive tests for radiation exposure, however, are yet to be developed. Microarray-based studies are identifying novel radiation responsive genes that have the potential to be adopted as biomarkers of human exposure to radiation [311]. The discussion focuses on biomarkers, which, in our opinion and according to the present knowledge, are better options to assess the effects of ionizing radiation in living matter.

### 4.1. Cytogenetics

Cytogenetic biomarkers help identify and quantify radiation-induced chromosomal aberrations. These biomarkers are currently used in human biomonitoring studies as parameters to assess the impact of environmental, occupational, and medical factors on genetic stability [312].

Ionizing radiation causes structural chromosome aberrations. Thus, the systematic measurement of cytogenetic endpoints helps in the assessment of risks linked to radiation exposure [313]. The mechanism of damage is not yet fully explored. A model of “breakage-first” followed by “reunion” is, however, widely accepted [314].

Differences in energy deposition and the quality of damage initially induced by low- and high-LET radiation affect the type and complexity of chromosome aberrations. In this regard, numerous cytogenetic markers and signatures on the basis of these differences have been proposed to discriminate exposure to radiation of varying qualities [314].

Radiation-induced structural aberrations in chromosomes can be found at any stage of the mitotic cycle. When cells are irradiated and after entering division, the occurrence of some changes in the surface properties of the chromosomes cause them to stick together. Most radiation-induced structural chromosomal aberrations are probably because of double-strand breaks [313,315]. This damage can either be repaired or remain as structural alterations identifiable as biomarkers.

Chromosomal aberrations are generally categorized into two different types: symmetrical (stable) and asymmetrical (unstable). Stability is defined as the ability of the rearranged chromosome to proceed through mitosis with no loss (or gain) of chromosome material [314]. Since Bender and Gooch [316] proposed the use of chromosomal aberrations for dose assessment in 1962, such observations have been used as a diagnostic tool used in radiation accidents, including mass-casualty incidents [317]. Current techniques for scoring chromosomal aberrations include a dicentric chromosome assay (DCA), a micronucleus assay or cytokinesis-block micronucleus assay (CBMN), a premature chromosome condensation assay (PCC), and fluorescence in situ hybridization (FISH) [318].

In the process of repair, a dicentric chromosome (DC, an unstable chromosome with two centromeres) is possible to be formed because of a mistake in the repair and the resulting abnormal chromosome replication [319]. Due to their unstable nature, DCs will decrease after exposure, with an approximately 2–3 year half-life [314]. DC estimates for a whole-body dose can be assessed on the basis of the peripheral blood lymphocyte DC frequency.

Micronuclei (MNs) are formed when intact chromosomes or their fragments are not properly segregated into daughter cell nuclei at the anaphase; however, following cell division, they rather stay in the cytoplasm. By means of any regular DNA dye, they can be pictured as small spherical objects. MNs are easy to score using, e.g., automated microscopy slide scanning and image analysis systems. The CBMN assay is a useful biomarker for individual radiosensitivity and environmental carcinogens [320].

PCC can be induced upon the fusion of mitotic cells or by treatment with chemicals such as calyculin A or okadaic acid. This assay, which identifies cells at the interphase with radiation-induced chromosomal aberrations, accurately discriminates between total- and partial-body exposure susceptibility [319]. PCC techniques can identify the condensation of chromosomes in a quiescent state and the cycling cells either by fusion with mitotic cells or by chemical treatment. PCC was reported to be the most helpful for assessing high-dose acute exposure to low-LET radiation [320].

Stable structural chromosome aberrations contain reciprocal translocations and insertions or inversions that cause monocentric rearranged chromosomes [314]. Whole-chromosome labeling by FISH has advanced substantially. It was initially able to just paint individual chromosomes with the same color, then capable of painting two or three different chromosomes the same or different colors, and, finally, had the ability to paint all human chromosomes in different colors [314]. The FISH-based quantification of reciprocal translocations reveals chromosomal aberrations that remain in peripheral blood lymphocytes for years and, thus, has the potential to be considered as past exposure biomarkers [320].

Despite the regular use of cytogenetic analysis in biodosimetry, there are several limitations associated with the analysis of cytogenetic data. Statistical problems generally occur due to either low numbers of aberrations leading to uncertainties or deviations in aberration-per-cell distributions. Moreover, the methodology also considers chromosome aberrations as a stable parameter, which leads to a merely deterministic estimate of the radiation dose.

### 4.2. Oxidative Stress-Induced Molecular Damage

As discussed above, oxidative stress is a common mediator of the damage induced by ionizing radiation. Some instances of the molecules are capable of being modified by excessive ROS in vivo, and a causal step in cell, organ, or system dysfunction involves DNA, lipids, proteins, and carbohydrates. The pros and cons associated with different oxidative stress biomarkers have been widely discussed (e.g., [321]). An ideal biomarker should have clinical applicability, stability under different storage conditions, and specificity, sensitivity, and reproducibility in measuring molecular alterations. To date, there is still an absence of a general agreement on validation, standardization, and reproducibility of methods for assays such as ROS in leukocytes and platelets via flow cytometry, markers based on the ROS-induced modifications of lipids, DNA, and proteins, enzyme activities controlling the cellular redox status, and the total antioxidant capacity of human bodily fluids [321]. Regarding the molecular damage caused by ionizing radiation (and keeping in mind the requirements described above), we selected biomarkers related to modifications of DNA, proteins, and lipids (see here below) (see, e.g., [322,323]).

Key examples are 8-hydroxy-2′-deoxyguanosine, an important mutagenic oxidative DNA lesion [324] protein carbonyl that can be caused by oxidative cleavage of the protein backbone, direct oxidation of different amino acids, or the binding of aldehydes produced from lipid peroxidation [325], and isoprostanes, which are prostaglandin-like compounds formed in vivo from the free-radical-catalyzed peroxidation of essential fatty acids [326].

All these oxidative stress-related molecular biomarkers are associated with the severity of radiation-induced tissue damage. However, the measurement of stable byproducts modified under the conditions of radiation-induced oxidative stress may not accurately reflect redox stress at the cell/tissue level. Many of these modifications are functionally silent, which represents a main limitation.

### 4.3. Immune and Inflammation Mediators

The way in which ionizing radiation and inflammation are connected is complex and multifactorial at the cellular and molecular levels. In general, lymphocytes (T cells, B cells, and NK) are among the most radiosensitive cells, followed by monocytes, macrophages, and antigen-presenting cells, specifically dendritic cells, which have higher radioresistance [327,328,329,330]. Ionizing radiation may also impact the immune system through the activation of cytokine cascades [331]. Chemokines, interferons, interleukins, lymphokines, and tumor necrosis factors, generated by immune and nonimmune cells, are released in excess due to irradiation, but their levels should return to baseline within a period of one to a few days (e.g., [332]). In this regard, the levels of pro-inflammatory (i.e., IL-1, IL-12, IL-18, TNF-α, IFN-γ, and GM-CSF) and anti-inflammatory cytokines (i.e., IL-1ra, IL-4, IL-6, IL-10, IL-11, IL-13, and TGF-β) may become critical for the response of an individual to irradiation [30] (Di Maggio 2015). For instance, early experiments demonstrated that administration of the anti-IL-1 receptor or anti-TNF-α antibody lowered the survival rate of irradiated CD2F1 mice [202]. Immortalized human keratinocytes underwent apoptosis and released IL-6 after γ irradiation [333]. Studies using the total RNA isolated from the mononuclear cells of different seemingly healthy adults exposed to low doses of ionizing radiation (0.18–49 mSv over a period of 11–13 years following Chernobyl disaster) revealed a transcriptional modulation of many cytokines, TNF-α/β, IL-1β, IL-2, IL-8, IL-10, IL-12β, M-CSF, and apoptosis-inducing receptors [334]. Radiation-induced cytokine release can promote cell apoptosis [335]. Studies on breast cancer cell lines demonstrated that 9 Gy and 23 Gy ionizing radiation in the first 72 h, caused the release of cytokines and growth factors possibly capable of influencing the outcome of the tumor [336], thus opening the prospect of evaluating cytokine profiles as a beneficial marker to modulate personalized radiotherapy. Although much work has been done on the relationship between cytokines and tumor radiosensitivity (e.g., [337]), cytokine levels in biological fluids should be further studied as a potential biomarker.

Although ionizing radiation is a known proinflammatory agent, the usefulness of this type of parameter in radiation biodosimetry can be compromised by many different conditions. For instance, a chronic inflammatory disease or any previous condition that may affect our immune response can act as confounding factors, thereby imposing strong limitations on assessing the impact of the inflammatory/immune response on predicting radiation exposure.

### 4.4. Gene Expression

Following a dose as low as 0.02 Gy, changes in gene expression in human cell lines start to show up. In peripheral blood lymphocytes, the same happens with doses as low as 0.2 Gy. Diverse genes are also elevated in vivo in mice 24 h after 0.2 Gy irradiation [338]. Exposure to ionizing radiation and other stresses leads to the activation of complex signal transduction pathways. The ATM/P53 pathway, MAPK cascades, and NF-κB activation, as well as signaling events initiated at the cell membrane and within the cytoplasm, are some examples of the noteworthy pathways responding to radiation. Thus, functional genomics may represent a good approach for radiation stress signaling [339]. However, further studies showed that very few genes are consistently upregulated by ionizing radiation—i.e., *GADD45*, *CDKN1A*, and the genes related to nucleotide excision repair pathway. All in all, the instant transcriptional reactions to ionizing radiation carry major repercussions for DNA repair, cell-cycle arrest, growth control, and cell signaling. Moreover, the transcriptional profile has not only a p53-independent component [340] but also a cell-specific p53-dependent component [341]. Indeed, the TP53 mutation status was proposed as a possible biomarker regarding precision radiation medicine [342]. Interestingly, some gene expression radiation signatures were revealed in a microarray analysis of spontaneous and post-Chernobyl thyroid cancers. One of these signatures could be related to the putative cause of the tumors and to a DNA repair pathway. The difference between the lymphocytes drawn from parents of children with retinoblastoma and those of parents of healthy children was also drawn by another gene expression signature, with the former being more radiosensitive [343]. Lu et al. discovered alterations in 29 genes involved in the cell cycle as a biomarker for predicting low doses of radiation exposure [344], and different studies have concluded that global gene expression alterations are crucial for the response of human cell to, at least, radiation exposure in low doses of ionizing radiation [345]. More recently, a global analysis identified a sum of 35 highly reproducible radiation-responsive genes. Most of these genes participate in reactions to DNA damage, cell proliferation, cell-cycle regulation, and DNA repair. The p53 signal pathway is the most enriched pathway [346]. Further studies have shown that, following exposure to 123IUdR, α-particles, and γ-rays with equi-effect doses/activities, 155, 316, and 982 genes were specially regulated, respectively. Four (*PPP1R14C*, *TNFAIP8L1*, *DNAJC1*, and *PRTFDC1*), one (*KLF10*), and one (*TNFAIP8L1*) genes were identified after exercising stringent requirements for candidate genes. These genes facilitated the reliable discrimination between γ- and 123IUdR exposure, γ- and α-radiation, and α- and 123IUdR exposure, respectively [347]. Further studies on radiation dose-dependent expressive genes in individuals exposed to external ionizing radiation suggested that the expression of several genes (i.e., *Slfn4*, *Itgb5*, *Smim3*, *Tmem40*, *Litaf*, *Gp1bb*, and *Cxx1c*) in whole blood is a sensitive and specific biomarker of radiation exposure and has the potential to be considered as a rapid and reliable prospective molecular biomarker in radiological emergencies [348]. Moreover, the expression of FDXR (ferredoxin reductase, a flavoprotein that transfers electrons from NADPH to mitochondrial cytochrome P450 enzymes) was significantly upregulated 24 h following radiotherapy in patients [349], thus suggesting FDXR (in circulating blood) as a potential in vivo radiation exposure biomarker.

Despite the clear advances in the development of gene expression signatures for practical radiation biodosimetry, many problems remain to be solved. Gene expression signatures are likely to be informative across a range of doses and throughout a window of time. However, many studies, such as those briefly described here, highlight the need to use multigene expression analysis to identify signature networks containing sets of functional pathways organized into a meta-network of gene clusters.

### 4.5. Gene Mutations

A main factor contributing to the negative biological effects of radiation in mammalian cells, such as chromosomal aberrations, mutations, and cell death, is the result of DNA damage in directly exposed cells—in other words, residual damage that has not been repaired by the metabolic processes in the exposed cell. These delayed effects can manifest in the unexposed progeny of irradiated cells for many cell divisions after the initial damage. This phenomenon is associated with the concept of radiation-induced genomic instability. Gene mutations are an endpoint that can be analyzed to evaluate this phenomenon.

Somatic mutations in the marker loci of hematopoietic stem cells in response to radiation can be monitored as biological indicators of the radiation dose. A large number of different reliable mutations for detecting radiation exposure have also been detected, for example, hemoglobin (Hb) and glycophorin A (gpa) variants in erythrocytes and mutations in HLA or hypoxanthine–guanine phosphoribosyl transferse (hprt) loci in T-lymphocytes [350]. The T-cell assay monitors mutations directly in circulating peripheral blood cells and facilitates mutant selection and the detection of the loss of a single allele. For instance, direct DNA sequencing showed that γ-radiation induced large deletions in the hprt locus of T-cells [351]. The main problem of HLA- and hprt-related assays is their required time. Nevertheless, this limitation could be overcome by, e.g., using an immunofluorescence assay to quantify hprt mutations in vivo.

### 4.6. Epigenetics

Chronic or acute irradiation can cause different epigenetic alterations, such as the formation of protein adducts that can affect epigenetic regulation, DNA methylation, histones, and signaling mechanisms controlling transcription factor expression [352,353]. The radiation-induced formation of electrophilic molecules causes the binding of these molecules to lysine groups in the histones, thus blocking the binding of these lysine groups to acetylated proteins and decreasing gene transcription activity [354]. Radiation-induced damage to the DNA can cause (a) DNA breaks and increased gene methylation, thereby producing gene silencing, or (b) hypomethylation due to methyl-cytosine hydroxylation, where the resulting 5-OH-methyl-cytosine is an epigenetic biomarker of DNA damage or hypomethylation due to ROS-induced glutathione oxidation, which affects the synthesis of *S*-adenosyl-methionine (essential substrate for DNA methyltransferases and histone methyltransferases) [355,356]. The differential regulatory functions of DNA methylation are dependent on the location of CpG-rich regions, which are present in the majority of repetitive elements [357]. Radiation-induced double-strand DNA breaks (DSBs), primarily repaired by polyADP (PAR) polymerases and chromodomain helicase DNA-binding protein 1-like, can also lead to gene upregulation [358]. Ionizing radiation also alters different signaling mechanisms, such as NF-κB signaling by inducing IkB degradation [359]. The loss of IGF2 cell imprinting is also a characteristic radiation-induced side effect, where the enhancer blocking element CCCTC-binding factor binds to the imprint control region of IGF2, thereby preventing the enhancers from allowing the transcription of the gene [360]. Moreover, the effects of in vivo radiation-induced bystanders are epigenetically regulated in a tissue-specific manner [361].

DNA methylation status, particularly its repetitive elements (which represent the largest methylated body of coding noncoding sequences in the genome), is a potential biomarker to study radiation-induced epigenetic alterations [362]. Interestingly, it has been demonstrated that the levels of expression in different tumor suppressor genes have different radiation threshold levels [363]. The study demonstrated the hypermethylation of the tumor suppressor gene *SOCS1* in the group receiving 30 rad. Moreover, genes related to the DNA damage response pathway (*GSTP1*, *ATM*, *DGKA*, *PARP1*, and *SIRT6*) were epigenetically inactivated in the groups receiving a carcinogen. Regarding proto-oncogene c-Myc, in the group with a low dose of infrared radiation (IR) (10 rad), DNA hypermethylation was detected [363], thus suggesting that, when considering radiation-induced epigenetic changes, organ-specific and radiation dose-dependent effects must be taken into account.

Furthermore, a phosphorylated form of the H2A histone family member X, γ-H2AX, forms when double-strand breaks appear, and it has been proposed to monitor radiation-induced DNA damage [364]. The main significance of the γ-H2AX assay relies on the fact that, following the exposure of cells to ionizing radiation, H2AX is γ-phosphorylated very quickly, reaching to half-maximal amounts at 13 min. Hundreds to several thousand γ-H2AX molecules are present per DNA double-strand break in mammals, at its maximum, which means 1030 min post irradiation [365]. A flow cytometry-based quantification of γ-H2AX has been recently validated [366]. Many reports on individuals medically exposed to low-dose radiation have demonstrated that the γ-H2AX assay is very sensitive. They also showed that foci after doses of 10–20 mGy are possible to be detected [320]. Moreover, the prospect of using hair bulbs for the γ-H2AX assay is extremely promising for epidemiological studies since the collection of hair bulbs is less invasive than blood or tissue collection.

Despite these facts, transgenerational changes of epigenetic markers following ionizing radiation exposure still play a poorly understood role. Currently, there are no reports of the transgenerational genome-wide effects of DNA methylation following ionizing radiation exposure in vertebrates. Therefore, a genome-wide analysis of DNA methylation (and of other epigenetic biomarkers) in a transgenerational set-up is necessary.

### 4.7. Metabolomics, Proteomics, and Lipidomics

Metabolomics involves the research on small-molecule metabolite (<1.5 kDa) profiles found in biological samples as chemical fingerprints of specific cellular processes [367], whereas metabonomics refers to the quantitative measurement of the dynamic metabolic response in a living system to pathophysiological stimuli (as radiation) or genetic modifications [368]. The Human Metabolome Database is the most extensive public metabolomic spectral database to date (www.hmdb.ca). Each type of cell and each sort of tissue are associated with an exclusive metabolic fingerprint linked to the information that are specific to the organ or tissue. This information has the potential to be impacted by a collection of various factors. Originally, metabolites were quantified by LC or GC coupled with MS and/or NMR spectroscopy [369,370,371]. Currently, bioinformatics tools are used to analyze data, identify associations, and establish metabolic signatures in response to, e.g., ionizing radiation damage and outcomes.

Early studies reported that, in mice, exposures of 3 and 8 Gy (γ radiation) produced completely different urine metabolomic phenotypes, where *N*-hexanoylglycine and beta-thymidine are urinary biomarkers of exposure to 3 and 8 Gy, 3-hydroxy-2-methylbenzoic acid 3-*O*-sulfate is raised in the urine of mice exposed to 3 but not 8 Gy, and taurine is high subsequent to 8 but not 3 Gy radiation [372]. Irradiated rats also showed abnormalities in phospholipid metabolism, which would potentially provide an substitute technique to analyze radiation exposure [373]. Deaminated purines and pyrimidines in the urine also identified a signature of urinary radiation metabolomic in mice exposed to ionizing radiation at sublethal doses [374]. The exposure of rats to 3 Gy (γ radiation) also revealed a considerable number of significantly upregulated urinary metabolites (i.e., glyoxylate, threonate, thymine, uracil, and *p*-cresol), as well as some other downregulated ones (i.e., citrate, 2-oxoglutarate, adipate, pimelate, suberate, and azelaate) due to radiation exposure [375]. Nine additional urinary biomarkers of γ radiation in rats (i.e., thymidine, 2′-deoxyuridine, 2′-deoxyxanthosine, *N*(1)-acetylspermidine, *N*-acetylglucosamine/galactosamine-6-sulfate, *N*-acetyltaurine, *N*-hexanoylglycine, taurine, and, tentatively, isethionic acid) were identified by ultra-performance LC-coupled time-of-flight MS [376]. More urinary markers (i.e., creatine, succinate, methylamine, citrate, 2-oxoglutarate, taurine, *N*-methyl-nicotinamide, hippurate, and choline) were identified in X-irradiated mice by NMR spectroscopy [377]. Further studies on nonhuman primates detected 13 potential biomarkers, i.e., *N*-acetyltaurine, isethionic acid, taurine, xanthine, hypoxanthine, uric acid, creatine, creatinine, tyrosol sulfate, 3-hydroxytyrosol sulfate, tyramine sulfate, *N*-acetylserotonin sulfate, and adipic acid (some of them were already detected in rodents) [378]. Studies on the urine of nonhuman primates are also available (e.g., [379]). Studies on the blood serum of γ-irradiatied rats also detected serum metabolites that were significantly altered due to radiation exposure. Upregulated metabolites were inositol, serine, lysine, glycine, threonine, and glycerol, whereas downregulated metabolites were isocitrate, gluconic acid, and stearic acid [380]. Studies on the internal exposure of mice to ^137^Cs showed comparable alteration in the urinary excretion levels of taurine and citrate to those seen with external-beam γ radiation, with no observed decrease in the levels of hexanoylglycine and *N*-acetylspermidine or any special impact on the levels of isovalerylglycine and tiglylglycine [381]. Moreover, at the lipidomic level, another study on mice also exposed to ^137^Cs revealed that fatty acids, for example, linoleic acid and palmitic acid, were present at lower levels in serum after radiation exposure, whereas in ^137^Cs-exposed mice serum, phosphatidylcholines were one of the most perturbed ions [382].

Studies in humans (subjects going through total-body irradiation at the Memorial Sloan Kettering Cancer Center (NYC) before hematopoietic stem cell transplantation at 4–6 h postirradiation (1.25 Gy) and 24 h (three fractions of 1.25 Gy each)) showed seven urinary markers with alterations between pre- and post-exposure, i.e., trimethyl-l-lysine and the carnitine conjugates acetylcarnitine, decanoylcarnitine, and octanoylcarnitine (related to the transport of fatty acids into the mitochondria), as well as hypoxanthine, xanthine, and uric acid (purine catabolism pathway’s final products) [383]. Noteworthy alterations in lipid metabolism, which alter all major lipid species, e.g., free fatty acids, glycerolipids, glycerophospholipids, and esterified sterols, were also reported in nonhuman primates who were exposed to ionizing radiation [384]. Further metabolic analyses of mice exposed to γ radiation (6 Gy) revealed a series of 67 biomarkers detected in radiosensitive tissues and biofluids (serum and urine). Among them, 3-methylglutarylcarnitine was identified as a unique metabolite in the liver, serum, and urine that has the potential of being used as a marker of early radiation response [385].

Recent studies have also revealed plasma-derived exosomal biomarkers of exposure to ionizing radiation in nonhuman primates, i.e., the enrichment of *N*-acyl-amino acids, fatty acid esters of hydroxyl fatty acids, glycolipids, and triglycerides in comparison with the nonirradiated control plasma metabolome composition [386].

Furthermore, in the metabolomic investigations on tissue injury in nonhuman primates exposed to γ radiation (7.2 Gy) over a 60 day time span, considerable metabolic alterations were found in the liver and kidney, as well as more moderate alterations in other tissue types [387].

In a very recent analysis searching for reliable radiation biomarkers, it was concluded that protein biomarkers from human plasma may be very valuable in this regard. Among a wide variety of potential markers, ferredoxin reductase was shown to be a very hopeful potential transcriptomic. More significantly, this biomarker has been ratified by some studies in humans at the protein level [388]. In addition, other differentially expressed protein biomarkers in rat blood plasma in response to γ irradiation have been identified, such as alpha-2-macroglobulin, chromogranin-A, and glutathione peroxidase 3 [389]. Moreover, several hundred dysregulated proteins have been identified in the serum samples of mice exposed to different doses of radiation (see, e.g., [390]), and plasma proteomic biomarkers are also capable of supplementing clinical signs and symptoms to evaluate the severity of hematopoietic acute radiation syndrome risk [391].

Radiation exposure triggers a complex network of molecular and cellular responses that impact the metabolic processes and alter the levels of metabolites. Although the above facts reflect only part of the available data in this vast field, it is remarkable that different potential radiation biomarkers have already been detected. Nevertheless, much more work is necessary to identify the markers or signatures related to specific organs or tissues, as well as genotypic differences, pathological conditions, the type and dose of radiation, and even radiation-induced syndromes. Nevertheless, the main challenges these potential biomarkers face is the fact that living tissues are dynamic and constantly adapting. This means that time and conditions may be critical and limiting.

### 4.8. MicroRNAs (miRNAs)

More than 15 years ago, sensitive serum miRNA biomarkers for radiation started to be identified [392], and serum miRNAs were also proposed as possible biomarkers of tumor radiation responses [393], as well as early indicators of survival after radiation-induced injury [394]. A very recent systematic review and meta-analysis, which screened 62 research studies, concluded that at least 28 miRNAs experience a considerable radiation-induced alteration in their expression. Moreover, even considering the differences among species (humans, rodent, and nonhuman primates), seven miRs (miR-150, miR-30a, miR-30c, miR-34a, miR-200b, miR-29a, and miR-29b) appear to alter their serum levels following radiation exposure [395]. Nevertheless, despite these advances, more work is necessary to validate the miRNA signatures that can reflect TBI or be organ-specific, syndrome-specific, and specific to the type of radiation or dose. For instance, studies by Port et al. [396] in baboons (subjected to partial or TBI, corresponding to an equivalent dose of 2.5 or 5 Gy) showed different miRNA signatures that may, indeed, be candidates for the early prediction of late-occurring hematologic ARS. Another interesting example is the study by Gao et al. [397], where, 2 weeks following thorax irradiation, an upregulation in four miRNAs, miR-144-5p, miR-144-3p, miR-142-5p, and miR-19a-3p, was identified in rat blood. The miRNA levels in the blood samples extracted from 0.5–10 Gy TBI mice, taken at 12 h to 7 days following exposure, were shown to be significantly predictive of the radiation dose [398]. Moreover, another study showed that a serum miRNA signature, whose detection is possible about 24 h following radiation exposure, has the ability of differentiating mice exposed to sublethal doses (6.5 Gy) from those exposed to lethal (8 Gy) doses of radiation [394]. The advantages of miRNAs include their stability and availability in different easily accessible fluids (serum, plasma, urine, and saliva), as well as in peripheral blood cells. Nevertheless, miRNA isolation requires the careful cooling of samples and use of RNase-free equipment to prevent RNAase-induced degradation. RT-PCR followed by quantitative PCR allows for the rapid quantification of miRNA expression. miRNAs can also be labeled with amine modified ribonucleotides and hybridized to antisense DNA oligonucleotide probes [399]. The miRNA microarray system is capable of detecting femtomole amounts of individual miRNAs from <1 μg of total RNA. Importantly, this array-based analysis can identify miRNAs expressed preferentially, in one or even in a few related tissues [399].

Despite all this, the results obtained in animal models are not easy to translate into clinical practice. Moreover, the mechanism through which miRNAs secrete into the serum is still unknown, as are the interactions between miRNAs and radioprotectors or radiomitigators. Furthermore, if miRNAs are linked to exosomes, they could participate in cell-to-cell transfer and, thus, play a role as a radiomitigating mechanism.

Table 2 schematically links all the potential biomarkers proposed along with their assays and methodologies, their validity for total-body or limited exposure to radiation, their applicability for ARS, and the possibilities for their analytical automation and incorporation into analytic networks.

## 5. Conclusions

Today, there remains a need for the development of effective countermeasures to protect people from the harmful effects of natural radiation, as well as medical exploratory techniques involving radiation, radiotherapy, and radiation-related incidents/accidents. As discussed in this review, many pharmacological agents with different targets and mechanisms are now subject to further research to prevent, alleviate, or treat ionizing radiation-induced toxicity. In this sense, the extensive range of agents that have the potential of acting as protectors and/or mitigators is on par with the complexity and diversity that various cell types and organs/tissues show in their responses to radiation. In this vast area of research, the main advantage of natural compounds is their high level of safety and toleration when compared to synthetic chemical compounds. Nevertheless, despite the obvious advantages of having ideal radioprotectors or radiomitigators, such agents have not yet been found. Moreover, various issues and questions still represent important barriers, such as the fact that radiation injury mechanisms are not yet completely identified, the potential in vivo toxicity associated with agents under development, why some molecules are potent protectants but rather poor mitigators, why some radioprotectors selectively target normal but not cancer cells, the fact that some radioprotectors also show anticancer properties, and finally the market size for these compounds relative to the investment required (research and development and approval). Furthermore, there is a need to effectively protect people against cosmic radiation if humanity wishes to expand beyond the earth.

Experimentally, candidate compounds assayed as potential protectors against ARS associated with a TBI are commonly tested in rodents. The studies in rodents adopted a 30 day survival as the major endpoint, while a 60 day survival was used in the case of nonhuman primates. However, the subsequent effects of all types of exposure (i.e., whole body, near whole body, and partial body), including among the survivors of ARS, are a rather neglected topic. Another poorly unexplored issue is how long a radioprotector or radiomitigator will work after radiation exposure. Neutrons, protons, and heavier ions comprise further uncharted water, as most of the research focuses on γ-radiation- or X-ray-induced effects. Importantly, the use of combinations of radioprotectors and/or radiomitigators is an almost unexplored field. On the basis of our present (although limited) knowledge and the variety of different mechanisms, selection of the right combination(s) could provide the sum of their protective effects and, eventually, synergisms. In our opinion, this option should be the subject of an in-depth investigation.

Furthermore, according to the available evidence, no single biomarker is sensitive and/or specific enough to evaluate the complexity of radiation-induced damage. Thus, at present, a multiparameter assessment method appears to be much more realistic. In this sense, the limitations and advantages of each biomarker discussed in this review must be considered.

The fundamental aim of this review was not only to describe and discuss the state of the art for radioprotectors, radiomitigators, and possible biomarkers that can evaluate the effects of radiation, but also to indicate the fundamental problems that must be solved and the most reasonable strategies to do so.

## Figures and Tables

**Figure 1 biomedicines-08-00461-f001:**
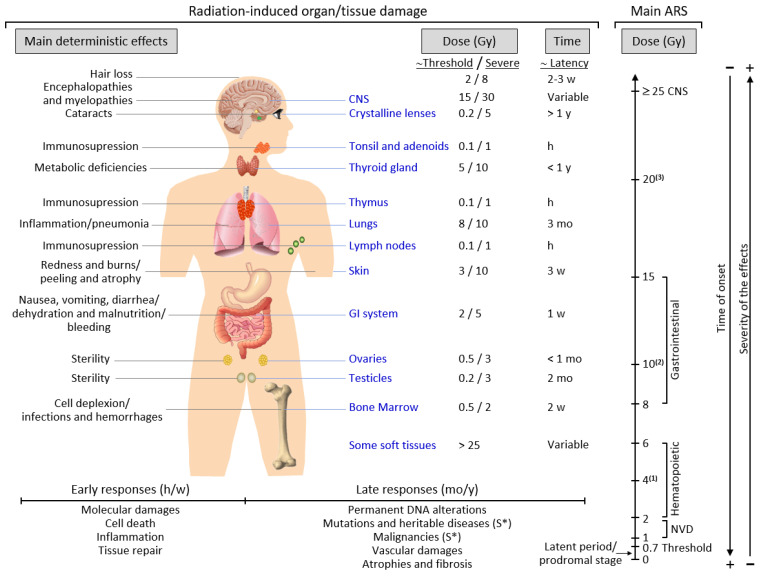
Consequences of exposure to ionizing radiation in organs and tissues are time- and dose-dependent. The main radiation-induced biological effects are displayed, while indicating the differences between the threshold doses and those that cause a severe effect. Time abbreviations: h, hours; w, weeks; mo, months; y, years. (S*) indicates a main stochastic effect. Acute radiation syndrome (ARS) may be difficult to differentiate from chronic radiation syndrome (CRS) since the dose threshold for the CRS is approximately 0.7–1 Gy (see the text for more details). (1) Possible death (approximately in 2 mo) due to bone marrow depletion. (2) Destruction of the intestinal lining, intestinal bleeding, possible death (1–2 w). (3) Cognitive impairment, convulsion, possible death (hours).

**Table 1 biomedicines-08-00461-t001:** Radioprotectors and radiomitigators tested in vivo and their radiomodifying mechanisms. Compounds are listed according to their main mechanism, although some have protecting and mitigating properties. Approval (+) means that a specific compound has been approved or is under clinical trials for use in humans in at least one medical indication. In this, we searched at www.fda.gov and www.clinicaltrials.gov. Only compounds marked as RP (radioprotector) or RM (radiomitigator) have been approved or a recognized as an investigational new drug for those specific uses. Due to the abundant literature, references included in this table only intend to be a representative example to lead the reader to a more extensive literature.

Compound	Main Mechanism(s) of Action	Approved	Ref.
**Radioprotectors**			
Thiol-containing molecules	Scavenging of free radicals, hydrogen donation, repair of damaged DNA, and Warburg-type- and HIF1α-dependent effects		
Amifostine (WR-2721)		RP	[47]
2-mercaptoethane sodium sulphonate		RP	[250]
Cysteine		+	[251]
*N*-Acetyl cysteine		+	[252]
Cysteamine		+	[253]
Cystamine			[254]
2-Mercaptoethylguanidine			[255]
Aminothiol PrC-210			[256]
Polyphenolic phytochemicals	Antioxidant and free-radical-scavenging activities, upregulation of physiological antioxidant defenses, anti-inflammatory properties		
Genistein		+	[60]
Resveratrol		+	[62]
Pterostilbene		+	[257]
Naringenin		+	[63]
Curcumin		+	[258]
Epigallocatechin-3-gallate		+	[259]
Epicatechin		+	[260]
Apigenin		+	[261]
Ellagic acid		+	[262]
Lutein		+	[263]
Nonpolyphenolic phytochemicals			
Caffeine	Antioxidant and anti-inflammatory properties	+	[73]
3,4-Methylenedioxyphenol	Protection of the hematopoietic and gastrointestinal systems		[74]
3,3′-Diindolylmethane	Stimulation of an ataxia-telangiectasia mutated-driven DNA damage response like response and NF-κB survival signaling	+	[75]
Vitamins			
Tocopherol and derivatives	Antioxidants and free radical scavengers, immunomodulators	+	[81]
δ- and ɣ-Tocotrienol	Antioxidants and free radical scavengers, immunomodulators, G-CSF increase, bone marrow and gastrointestinal protection	+	[83]
Ascorbic acid	Antioxidant and free-radical scavenger, gastrointestinal protection	+	[79]
β-Carotene	Antimutagenic and antioxidant activity	+	[264]
Retinol	Anti-inflammatory, MMP inhibitor, free-radical scavenger	+	[76]
α-Lipoic acid	Antioxidant and free-radical scavenger	+	[265]
Oligoelements			
Zinc	Development and functions of different immune cells, inhibition of NF-κB signaling, promotes DNA repair, stabilizes sulfhydryls, SOD1 activity	+	[266]
Copper		+	[267]
Manganese	SOD2 activity	+	[268]
Mn-porphyrin derivatives	Free-radical scavengers	+	[269]
Selenium and sodium selenite	Increases the activity DNA glycosylases, GSH peroxidase activity	+	[91]
Selenomethionine	Upregulates p53-mediated base excision repair pathways	+	[90]
Selenopropionic acid	Inhibition of DNA strand break formation		[92]
Selenocysteine	Forms part of the catalytic center of GSH peroxidases		[93]
Cerium oxide nanoparticles	Free radical scavenging activity, SOD2 regulation		[270]
Superoxide dismutase (SOD)	Antioxidant		
SOD2		+	[97]
AEOL 10150 (Mn porphyrin SOD mimic)			[271]
AAV2-Mn-SOD-hrGFP			[100]
SOD2 gene-modified mesenchymal stem cells			[272]
BMX-001 (a SOD2 mimic)		+	[103]
EUK-134, EUK-189, EUK-207 (synthetic MN complexes SOD/catalase mimetics)		+	[273]
Ex-Rad (recilisib sodium)	Upregulation of PI3-kinase/AKT pathways, hematopoietic and gastrointestinal protection		[107]
Nitrosides			
Tempol	Antioxidant activity, prevention of chromosomal aberrations	+	[113]
JP4-039 (mitochondria-targeted GS-nitroxide)	Enhances intestinal barrier and stem-cell recovery		[274]
Hormones and hormone analogues			
Melatonin	Antioxidant and anti-inflammatory activity	+	[115]
Indraline (alpha-adrenomimetic)	Increase in DNA and protein biosynthesis and ribonucleotide reductase		[122]
Androstenetriol	Stimulate the innate immune system		[275]
Antibiotics			
Streptomycin, kanamycin, neomycin, gentamicin	Protect hematopoietic system	+	[125]
Tetracyclines	Protect hematopoietic stem/progenitor cell population, free-radical-scavenging activity, protect DNA	+	[1]
Fluoroquinolones, minocyclin	Antioxidant and free-radical-scavenging activity, protect hematopoietic system	+	[276]
Adenosine receptor agonists			
AMP and dipyridamole	Systemic vasodilation, hypotension, hypoxia and elevation of cAMPin sensitive cells), hematopoiesis stimulation, favors DNA stability	+	[130]
IB-MECA	Hematopoiesis stimulation	+	[131]
G-CSF	Hematopoiesis stimulation	+	[277]
DNA-binding molecules	Favor stability and prevent damage of DNA		
Hoechst			[140]
Methylproamine			[141]
Netropsin			[142]
Pentamidine		+	[144]
Other			
Anethole and anethole ditholethione (unsaturated ether related to lignols)	Anti-inflammatory (inhibit TNF-induced NF-κB activation), increase GSH		[278]
LY294002 and PX-867	PI3K inhibitors that decrease radiation-induced DNA damage		[279]
SB-415286 (a specific inhibitor of GSK-3β)	Protection for radiation-induced necrosis in brain and the intestine		[280]
C60(OH) (a polyhydroxylated fullerene derivative)	Protects from radiation-induced immune and mitochondrial dysfunction		[281]
**Radiomitigators**			
l-Glutamine	Reducing the incidence of gastrointestinal, neurological, and cardiac complications, accelerates healing of the irradiated bowel	+	[147]
Probiotics	Mitigate the radiation-induced gastrointestinal damage, reduce diarrheal symptoms		[155]
2nd-generation producing IL22	Prevents loss of intestinal crypt cells (principally Leu-rich repeat-containing G-protein-coupled receptor 5 stem cells)		[158]
ACE inhibitors and angiotensin II receptor blockers			
Perindopril	Increases bone marrow cellularity and the hematopoietic progenitors CFU-GM, BFU-E, and CFU-Meg, mitigates the hematopoietic toxicity	+	[159]
Captopril	Upregulates the hematopoietic progenitor cell cycle		[160]
Angiotensin receptor blockers	Interact with the angiotensin AT1 receptor to specifically release PGE2	+	[164]
Statins	Reduce the mRNA expression of pro-inflammatory and pro-fibrotic cytokines, accelerate the repair of DNA double-strand breaks and mitigate DNA damage		[168]
Simvastatin	Reduces cardiac dysfunction and capsular fibrosis	+	[171]
Somatostatin analogues			
Octreotide	Ameliorates acute and delayed intestinal radiation injury	RM	[177]
SOM230 (pasireotide)	Ameliorates injury of the intestinal mucosa	+	[180]
Immunomodulators			
β-Glucan	Stimulates hematopoiesis	+	[189]
5-Androstenediol	Stimulates hematopoiesis and the innate immune system, inhibits the inflammasome-mediated pyroptosis	RM	[196]
Peptidoglycan	Ameliorates intestinal and bone marrow damages		[53]
CBLB502	Activates TLR5 and triggers NF-κB signaling, mobilizing an innate immune response	+	[197]
CBLB613	Protects against the hematopoietic syndrome		[248]
Antibiotics and synthetic trehalose dicornomycolate	Prevent sepsis following irradiation	+	[199]
Cytokines			
G-CSF and filgrastim, GM-CSF, and sargramostim	Upregulate the production of neutrophils within the bone marrow and minimizes the radiation-induced hematopoietic syndrome	RM	[197]
TGF-β	Decreases pulmonary fibrosis	+	[204]
Palifermin (recombinant N-terminal truncated KGF)	Stimulates epithelial cell proliferation, differentiation, and upregulates cytoprotective mechanisms	+	[206]
RSPO1	Efficacy against radiation-induced mucositis, promotes gastrointestinal epithelial cell proliferation	RM	[207]
AG1024 (an IGF-1R inhibitor)	Prevents radiation-induced endothelial cell senescence	+	[282]
Bevacizumab (anti-VEGF antibody)	Ameliorates the edema associated with radiation necrosis	+	[211]
Octadecenyl thiophosphate (a synthetic mimic of the GF-like mediator lysophosphatidic acid)	Mitigates the hematopoietic and gastrointestinal ARS	RM	[283]
UTL-5g (TNF-α inhibitor)	Protects in the acute phase of radiation-induced liver injury	+	[284]
IL-1	Protects the hematopoietic stem cells	+	[203]
IL-12 and rhIL12 (HemaMax)	Promote hematopoiesis and recovery of immune functions	RM	[285]
IGF-1	Protection of intestinal stem cells	+	[286]
CDX-301 (recombinant human protein form of the Fms-related tyrosine kinase 3 ligand)	Stimulates expansion and differentiation of hematopoieticprogenitor and stem cells	+	[287]
FGF-peptide	Improves barrier function and proliferation in human keratinocytes, promotes stem cell self-renewal	+	[288]
NSAIDs			
Meloxicam (selective COX2 inhibitor)	Stimulates hematopoiesis	+	[218]
Acetylsalicylic acid	Ameliorates radiation-induced kidney and lung damage, and reduces chromosomal aberrations	+	[221]
Bio 300 (genistein nanoparticles)	Reduces the formation of collagen-rich lesions in irradiated lungs	RM	[223]
ABC294640 (inhibitor of sphingosine kinase-2)	Reduces radiation-induced gastrointestinal inflammation	+	[230]
Cell therapy			
MSCs	Ameliorate radiation-induced injury in liver and brain, promote healing of the irradiated skin, and mitigate radiation-induced gastrointestinal and hematopoietic syndromes	RM	[245]
Bone marrow stromal cell transplantation	Restitutes irradiated intestinal stem cells niche and mitigate radiation-induced gastrointestinal syndrome	+	[246]
Transplantation of myeloid progenitors and megakaryocyte/erythrocyte-restricted progenitors	Protection against the radiation-induced hematopoietic syndrome	+	[289]
PLX-R18 (allogenic placenta-derived stromal cells)	Maintenance and renewal of hematopoietic progenitor cells	RM	[290]
CLT-008 (human myeloid progenitor cells)	Promote neutrophil and platelet recovery, and human myeloidprogenitor cell persistence	RM	[289]
Others			
ALXN4100TPO (thrombopoietin receptor agonist)	Stimulates platelet production	RM	[291]
2A2 (an anti-ceramide antibody)	Prevents ceramide-induced endothelial cell apoptosis within the mucosal microvascular network		[292]
Rapamycin (a specific inhibitor of mTOR)	Mitigates radiation-induced pulmonary fibrosis, and protects hematopoietic cells and the liver	RM	[293]
TP508 (rusalatide acetate)	Mitigates effects of radiation by restoring endothelial function, andrecovery of bone marrow stem cells and stem cells in intestinal and colonic crypts	RM	[294]
Granistron, ondansetron (serotonin 5-HT3 receptor antagonists)	Mitigate radiation-induced nausea and vomiting	RM	[295]
Poly *N*-(acetyl, arginyl) glucosamine	Mitigates the radiation-induced proctitis, reduces inflammation andlesions in radiation-induced oral mucositis	+	[13]
Metformin	Targets endogenous ROS production within cells, and enhancesDNA repair capacity	RM	[296]

**Table 2 biomedicines-08-00461-t002:** Biomarkers of ionizing radiation-induced damage. Different assays and methodology are recommended for each type of biomarker. Dose range and timing are also indicated. TBI, total-body irradiation; PBI, partial-body irradiation; PBL, peripheral blood lymphocytes; TBD, to be determined. All other abbreviations, as well as references for each specific biomarker, can be found in Section 4.

Biomarkers	Assay	Methodology	Dose Range	Time Window after Exposure	TBI	PBI	Applicable for ARS Scoring	Automation	Recommended for Networks
**Cytogenetics**	DCA	Fluorescence + Giemsa staining	0.1–5 Gy	Before renewal of PBL	yes	yes	yes	yes	yes
	CBMN	Fluorescence + DAPI	0.2–4 Gy but with limited sensitivity at dose <1 Gy	In lymphocytes: before renewal of PBL In reticulocytes: not yet established	yes	yes	TBD	yes	yes
	PCC	PEG-induced fusion of G0 lymphocytes with mitotic CHO cells	PCC fragments: 0.2–20 Gy PCC rings: 1–20 Gy	PCC fragments: ideally immediately after exposure PCC rings: before renewal of PBL	yes	limited	no	no	yes
	FISH	Cell cycle control for FISH painting using the FpG method	0.25–4 Gy	Years	yes	no	TBD	no	yes
**Oxidative stress**	8-hydroxy-2′-deoxyguanosine, isoprostanes and protein carbonyls	UPLC–MS/MS, ELISA	>0.1 Gy	1–2 h	yes	yes	no	no	yes
**Immune and inflammatory mediators**	IL-1, IL-1ra, IL-1β, IL-2, IL-4, IL-6, IL-8, IL-10, IL-11, IL-12, IL-12β, IL-13, IL-18, TNF-α, TNF-a/b, IFN-γ, TGF-β, GM-CSF, M-CSF	ELISA	>1.2 mGy	Within 24 h	yes	no	yes	yes	yes
**Gene expression and mutations**	Gene expression (ATM/P53 pathway, MAPK cascades, NF-κB activation, *GADD45*, *CDKN1A*, genes associated with the nucleotide excision repair pathway, *TP53, PPP1R14C, TNFAIP8L1, DNAJC1, PRTFDC1, KLF10, TNFAIP8L1, Slfn4, Itgb5, Smim3, Tmem40, Litaf, Gp1bb, Cxx1c, FDXR*)	RT-PCR	0.1–10 Gy	Hours–days	yes	yes	yes	yes	yes
	gpa	ELISA	>1 Gy	Years					
	hprt	Direct DNA sequencing or immunofluorescence	>90 mGy	Months					
**Epigenetics**	Gene methylation and repetitive elements	HPLC-UV, LC–MS/MS, ELISA, qPCR or PCR and sequencing	0.1–10 Gy	Hours–months	yes	TBD	TBD	yes	yes
	γ-H2AX	Immunofluorescence staining, flow cytometry, electro-chemiluminescence	0.01–8 Gy	Minutes–days	yes	yes	TBD	yes	yes
**Omics**	Metabolomics: urine metabolites (*N*-hexanoylglycine and beta-thymidine, glyoxylate, threonate, thymine, uracil, *p*-cresol, citrate, 2-oxoglutarate, adipate, pimelate, suberate, azelaate, thymidine, 2′-deoxyuridine, 2′-deoxyxanthosine, *N*(1)-acetylspermidine, *N*-acetylglucosamine/galactosamine-6-sulfate, *N*-acetyltaurine, *N*-hexanoylglycine, taurine, creatine, succinate, methylamine, citrate, 2-oxoglutarate, *N*-methyl-nicotinamide, hippurate, choline, xanthine, hypoxanthine, uric acid, creatine, creatinine), blood serum metabolites (inositol, serine, lysine, glycine, threonine, glycerol, isocitrate, gluconic acid, stearic acid, methylglutarylcarnitine)	LC or GC coupled with MS and/or NMR spectroscopy	0.5–10 Gy	Hours–months	yes	no	yes	yes	yes
	Proteomics: plasma (ferredoxin reductase, alpha-2-macroglobulin, chromogranin-A, glutathione peroxidase 3)	HPLC–LC–MS/MS, SELDI-TOF-MS,	1–14 Gy	10 min–40 weeks post-irradiation	yes	no	TBD	yes	yes
		DIGE, MALDI-TOF/TOF, Western blot, tissue arrays							
	Lipidomics: blood serum (as linoleic acid, palmitic acid, phosphatidylcholines, glycerolipids, glycerophospholipids and esterified sterols)	LC or GC coupled with MS and/or NMR spectroscopy	6–10 Gy	2–3 days post-irradiation	yes	no	TBD	yes	yes
**microRNAs (miRNAs)**	Blood serum (miR-150, miR-30a, miR-30c, miR-34a, miR-200b, miR-29a, miR-29b, miR-144-5p, miR-144-3p, miR-142-5p, miR-19a-3p)	Microarrays, RT-PCR, nanostring nCounter technology, NGS	0.5–12 Gy	Hours–months	yes	no	no	yes	yes

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
