# Peer review of "Radioprotection and Radiomitigation: From the Bench to Clinical Practice"

_biomedicines, 2020, doi:10.3390/biomedicines8110461_

Round 1

Reviewer 1 Report

The authors have posted blockbuster reviews of radioporotection and radiomitigation, covering everything from basic research to clinical trials.

In some places, I don't feel any particular problem with the description, except that the space between words seems to disappear. Please check this again as a whole.

The volume is so large that it may be better to divide it into two parts (two papers), the first half and the second half, but it is considered to be at the discretion of the editor or the editorial department.

Author Response

Editor

Biomedicines

Manuscript ID: biomedicines-974005

Radioprotection and radiomitigation: from the bench to clinical practice

Oct. 27, 2020.

Dear Editor,

I sincerely want to thank both referees their thoughtful reviewing of the manuscript. Please find here below a point-by-point response to the criticisms/suggestions raised in the editorial report.

There are two copies of the revised manuscript uploaded into the journal’s system. One clean copy, and another copy (tracked changes) where all changes introduced have been highlighted in YELLOW color. All PAGES mentioned in this point-by-point response refer to the copy with TRACKED CHANGES.

We hope this revised version is now acceptable for publication in Biomedicines.

Yours, cordially,

Alegría Montoro.

Corresponding author.

POINT-BY-POINT RESPONSE

REFEREE 1.

  1. In some places, I don't feel any particular problem with the description, except that the space between words seems to disappear. Please check this again as a whole.

Answer: We have checked this issue in the manuscript. Furthermore, our manuscript has undergone English language editing by MDPI (see here below):

  1. The volume is so large that it may be better to divide it into two parts (two papers), the first half and the second half, but it is considered to be at the discretion of the editor or the editorial department.

Answer: unless required by the Editor, we will keep presenting this contribution as a single full paper. But, in this regard, we will bow to the editor’s decision.

Reviewer 2 Report

The text throughout the manuscript should be reviewed because there are together words, mistakes in spaces between words, etc.

Page 2, Introduction:

  • The reference [1] does not correspond with the information in the text.
  • In paragraph 4, a text about the contribution of radiotherapy to the collective effective dose could be included. For example, neither the 2009 or 2019 NCRP report incorporated doses from radiotherapy into calculated population dose exposures, as the assessment of effective dose for the population undergoing radiotherapy is more complex than that for other medical radiation exposures (https://doi.org/10.1016/j.jacr.2020.02.002).

Page 3, section 2.1:

  • More explanation should be presented about RBE (such as definition and relationship between RBE and LET). The reference [1] does not correspond with the information in the text.
  • The text about RBE, adsorbed dose (Gy) and equivalent/effective dose (Sv) should be rewritten. Sievert is not the biological effect.
  • Alpha, beta, gamma and neutron radiation are commented. Charged particles (such as protons and carbon ions) should be also commented because are used in hadrontherapy.

Page 4, section 2.3:

  • In late responses, add information about stochastic effects (cancer)

Page 5, figure 1:

  • The Figure and footnote are incomplete.

Page 19, table 1:

  • Review mistakes in the text such as C60(OH)+62:7724 and Ace inhibitors and angiotensin II receptor blockers+82:120
  • The table should be redesign because sometimes is difficult to align compound number, mechanism of action and ref.; for example, include sublines between compounds, etc.

Page 25, section 3.3.7:

  • References on decorporation of actinides with chelating agents should be added; e.g., DOI:10.1016/j.apradiso.2005.01.005 and DOI:10.1016/j.crci.2007.01.015

Page 36, table 2:

  • In the header row: change “Metodology” for “Methodology”; “Automation” and “Recommended” are underlined??
  • The last row in this page only shows Methodology??

Author Response

Editor

Biomedicines

Manuscript ID: biomedicines-974005

Radioprotection and radiomitigation: from the bench to clinical practice

Oct. 27, 2020.

Dear Editor,

I sincerely want to thank both referees their thoughtful reviewing of the manuscript. Please find here below a point-by-point response to the criticisms/suggestions raised in the editorial report.

There are two copies of the revised manuscript uploaded into the journal’s system. One clean copy, and another copy (tracked changes) where all changes introduced have been highlighted in YELLOW color. All PAGES mentioned in this point-by-point response refer to the copy with TRACKED CHANGES.

We hope this revised version is now acceptable for publication in Biomedicines.

Yours, cordially,

Alegría Montoro.

Corresponding author.

POINT-BY-POINT RESPONSE

REFEREE 2.

  1. The text throughout the manuscript should be reviewed because there are together words, mistakes in spaces between words, etc.

Answer: We have checked this issue through the entire manuscript. In addition, our manuscript has undergone English language editing by MDPI.

  1. Page 2, Introduction. The reference [1] does not correspond with the information in the text.

Answer: OK. Ref. 1 has been changed to Zakariya NI, Kahn MTE. Benefits and Biological Effects of Ionizing Radiation. Sch. Acad. J. Biosci., 2014; 2(9): 583-591.

  1. Page 2, Introduction. In paragraph 4, a text about the contribution of radiotherapy to the collective effective dose could be included. For example, neither the 2009 or 2019 NCRP report incorporated doses from radiotherapy into calculated population dose exposures, as the assessment of effective dose for the population undergoing radiotherapy is more complex than that for other medical radiation exposures (https://doi.org/10.1016/j.jacr.2020.02.002).

Answer: The contribution of radiotherapy to the collective effective dose received by the general population has been a matter of controversy. For instance, the NCRP Report No. 184 committee elected to not incorporate radiation dose from radiotherapy into its calculated population dose exposures (Milano MT, Mahesh M, Mettler FA, Elee J, Vetter RJ. Patient Radiation Exposure: Imaging During Radiation Oncology Procedures: Executive Summary of NCRP Report No. 184.  J Am Coll Radiol. 2020 Sep;17(9):1176-1182). The basic argument raised in this report was that assessment of collective effective dose for the population undergoing radiotherapy is more complex than that for other medical radiation exposures. However, the large number of patients receiving courses of radiotherapy, which frequently include different types of imaging, raises the question of whether radiotherapy-derived doses should also be considered for calculations. Nevertheless, in most cases radiotherapy is applied to an organ or specific part of the body. Moreover, heterogeneity in radiotherapy regarding the individual dose distribution adds complexity to the actual contribution to the collective effective dose. These facts and the limited number of patients receiving radiotherapy, compared to those subjected to complex radiology or CST, may explain the position expressed in the above mentioned NCRP Report. See page  2, last paragraph; and page 3, 1st paragraph.

  1. Page 3, section 2.1. More explanation should be presented about RBE (such as definition and relationship between RBE and LET). The reference [1] does not correspond with the information in the text.

Answer: The RBE is defined as the ratio between an absorbed standard dose of radiation (typically X), and the absorbed dose of any other type of radiation that causes the same amount of biological damage. In many cases, the biological effect of radiation increases in proportion to the increase in LET. Radiations commonly used to assess RBE are low LET X or gamma, for which RBE is 1.0. However, when evaluating some biological effects caused by high LET radiation (such as fast neutrons), the RBE can vary widely, from about 3 to more than 100 depending on the cellular or tissue effect considered. For example, higher RBEs for neutron radiation are associated with high LET effects, which are directly linked with protons released by collisions of these neutrons with hydrogen nuclei (see e.g. Sørensen BS, Overgaard J, Bassler N. In vitro RBE-LET dependence for multiple particle types. Acta Oncol. 2011 Aug;50(6):757-62). Consequently, doses should be evaluated in terms of absorbed dose (in Gy), and when high-LET radiations are involved the absorbed dose must be correlated with an appropriated RBE. See page 3, section 2.1.

As indicated above (your question #2), Ref. 1 has been changed.

  1. The text about RBE, adsorbed dose (Gy) and equivalent/effective dose (Sv) should be rewritten. Sievert is not the biological effect.

Answer: OK. See also above. One Sievert (Sv) is defined as the amount of radiation roughly equivalent in biological effectiveness to one Gy (or 100 rads) of gamma radiation. Conventionally, the Sv is not used for high dose rates of radiation that produce deterministic effects, which is the severity of acute tissue damage that is certain to happen, such as acute radiation syndrome; these effects are compared to the physical quantity absorbed dose measured by the unit Gy. The sievert is used for radiation dose quantities such as equivalent dose and effective dose, which represent the risk of external radiation from sources outside the body, and committed dose which represents the risk of internal irradiation due to inhaled or ingested radioactive substances. The Sv is intended to represent the stochastic health risk, which for radiation dose assessment is defined as the probability of radiation-induced cancer and genetic damage.

To avoid misleading interpretations, the following text has been added to the revised version: The Sievert (Sv), defined as the corresponding biological effect of the deposit of one joule of radiation energy in one kg of human tissue (www.icpr.org), is used to evaluate the biological effect of low doses of ionizing radiation representing the risk of external radiation from sources outside the body, and those representing the risk of internal irradiation due to accidentally inhaled or ingested radioactive substances. The Sv helps to value the stochastic health risk, which represents the probability of radiation-induced cancer and genetic damages. See page3, bottom.

  1. Alpha, beta, gamma and neutron radiation are commented. Charged particles (such as protons and carbon ions) should be also commented because are used in hadrontherapy.

Page 4, section 2.3:

Answer: OK. We have added the following comments to the revised version:

Proton and carbon ion therapy are two types of hadron therapy which have been increasingly used in recent years for cancer treatment.

Proton therapy uses a beam of protons to irradiate tissues, most often as a type of cancer therapy. Its main advantage is that the dose is deposited over a narrow range of depth, which results in minimal entry, exit, or scattered radiation dose to healthy nearby tissues (De Marzi L, Patriarca A, Scher N, Thariat J, Vidal M. Exploiting the full potential of proton therapy: An update on the specifics and innovations towards spatial or temporal optimisation of dose delivery. Cancer Radiother. 2020 Oct;24(6-7):691-698).

Carbon ions exhibit a characteristic energy distribution in depth, known as the “Bragg Peak,” where low levels of energy are deposited in tissues proximal to the target, and the majority of energy is released in the target. Its main advantage is that may allow dose escalation to tumors while reducing radiation dose to adjacent normal tissues (Malouff TD, et al. Carbon Ion Therapy: A Modern Review of an Emerging Technology. Front Oncol. 2020 Feb 4;10:82). See page 4, 5th paragraph.

7.Page 4, section 2.3.  In late responses, add information about stochastic effects (cancer)

Answer: information related stochastic effects has been included in this section.

In the 1st paragraph (page 4, bottom), under B. Stochastic (random), i.e., cell mutation-associated pathologies and heritable diseases following moderate, we have included “cancer” and “heritable effects”.

In page 5, 3rd paragraph, right after nutrient deprivation [35], the following statement has been added: Stochastic effects likely derive from an injury to a single cell or a small number of cells. Cancer induction is the most important somatic late-effect of low-dose radiation exposure.

  1. Page 5, figure 1. The Figure and footnote are incomplete.

Answer: Fig. 1 (page 6) has been revised and completed.

  1. Page 19, table 1. Review mistakes in the text such as C60(OH)+62:7724 and Ace inhibitors and angiotensin II receptor blockers+82:120

Answer: Yes, you are right. C60(OH)+62:7724 has been changed to C60(OH)24; and

+82:120  has been removed from Ace inhibitors and angiotensin II receptor blockers.

  1. Table 1 should be redesign because sometimes is difficult to align compound number, mechanism of action and ref.; for example, include sublines between compounds, etc.

Answer: We have included some extra space and separation lines in order to facilitate the reader a better alignment of each compound, its mechanism of action and the indicated ref.

  1. Page 25, section 3.3.7. References on decorporation of actinides with chelating agents should be added; e.g., DOI:10.1016/j.apradiso.2005.01.005 and DOI:10.1016/j.crci.2007.01.015

Answer: Both articles have been added. Please see page 27, 1st paragraph.

  1. Page 36, table 2. In the header row: change “Metodology” for “Methodology”; “Automation” and “Recommended” are underlined??

Answer: In Table 2 (page 35), the mistake regarding Methodology has been corrected. Automation and Recommended are not underlined.

  1. Table 2. The last row in this page only shows Methodology??

Answer: There was just an extra line which has been eliminated.